# The Geometry of Adversarial Subspaces

## Abstract

Artificial neural networks (ANNs) are constructed using well-understood mathematical operations, and yet their high-dimensional, non-linear, and compositional nature has hindered our ability to provide an intuitive description of how and why they produce any particular output. A striking example of this lack of understanding is our inability to design networks that are robust to adversarial input perturbations, which are often imperceptible to a human observer but cause significant undesirable changes in the network's response. The primary contribution of this work is to further our understanding of the decision boundary geometry of ANN classifiers by utilizing such adversarial perturbations. For this purpose, we define adversarial subspaces, which are spanned by orthogonal directions of minimal perturbation to the decision boundary from any given input sample. We find that the decision boundary lies close to input samples in a large subspace, where the distance to the boundary grows smoothly and sub-linearly as one increases the dimensionality of the subspace. We undertake analysis to characterize the geometry of the boundary, which is more curved within the adversarial subspace than within a random subspace of equal dimensionality. To date, the most widely used defense against test-time adversarial attacks is adversarial training, where one incorporates adversarial attacks into the training procedure. Using our analysis, we provide new insight into the consequences of adversarial training by quantifying the increase in boundary distance within adversarial subspaces, the redistribution of proximal class labels, and the decrease in boundary curvature.

## 1 Introduction

Artificial neural networks (ANNs) have been highly performant on common machine learning tasks, but their application space is limited by their susceptibility to adversarial attacks (Szegedy et al., 2014), which underscores a general lack of understanding for how and why they make their decisions. Examples produced by adversarial attacks are a worst-case demonstration of an ANN's inability to gracefully cope with identity-preserving shifts or distortions of its inputs. To construct one, an adversary must perturb the input in a small but specific way such that the output of the network changes significantly. Adversarial perturbations are defined to have minimal length, resulting in inputs that are as close to the network's decision boundary as possible. Here, we present a method that uses an untargeted adversarial attack objective to find the multi-dimensional subspace where an ANN's decision boundary is closest to any given input sample. The resulting *adversarial subspace* allows us to visualize and understand ANN decision boundaries in terms of their curvature, their shape, and their proximity to the input.

One existing technique to explain ANN classifications has relied on two-dimensional visualizations of the decision surface, although it suffers from a number of shortcomings, most notably high-dimensional points resolving to the same projected location, distortions of distances, and a lack of invertibility from the two-dimensional space back to the high-dimensional decision space (Féraud & Clérot, 2002; Rauber et al., 2016; Rodrigues et al., 2019). A popular alternative strategy is to investigate the decision boundary in the high-dimensional input space itself, early attempts of which (Golland, 2001; Baehrens et al., 2010) lead the way to the modern interest in adversarial attacks (Biggio et al., 2013; Szegedy et al., 2014). For example, one can measure the distance to the decision boundary for random perturbation

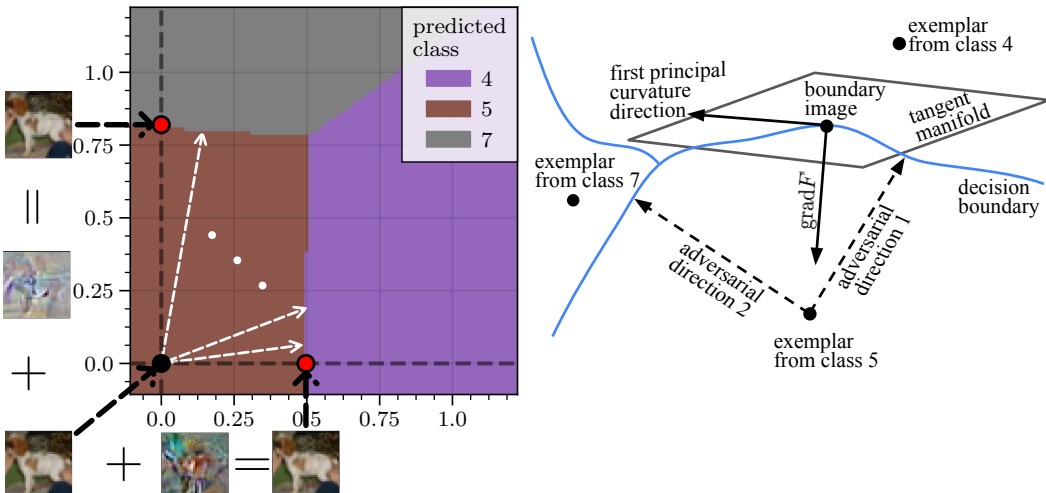

Figure 1: **An adversarial subspace**. (Left) For any given input example, we define a subspace where the boundary is closest to the input. The white arrows depict our sampling scheme, which uniformly samples directions within the adversarial subspace where we measure the distance to the decision boundary. The perturbation images are rescaled for visualization, but their $\ell_2$ length is indicated on the axes. Each color corresponds to the class categories for deer (class 4), dogs (class 5), and horses (class 7). This visualization is for an adversarially trained ResNet-50. (Right) We then measure the curvature of the decision boundary, both in the high-dimensional input space and in the adversarial subspace.

directions away from adversarial examples (Tabacof & Valle, 2016; He et al., 2018), or for orthogonal directions which are proximal to an initial adversarial example (Tramèr et al., 2017). Alternatively, much work has demonstrated the utility of visualizing two-dimensional cross-sections of the decision space between adversarial examples and random orthogonal directions (Warde-Farley & Goodfellow, 2016; Yu et al., 2019a) or orthogonal test set examples (Swirszcz et al., 2019). Unlike all of the existing approaches to characterize decision boundaries in the input space, which rely on at most one adversarial direction per input, we propose jointly defining a subspace of adversarial directions. The resulting analysis leads to a general explanation of classifier decision boundaries, as well as adversarial susceptibility, in the subspace where the boundary is closest to the input example.

As a further demonstration of the utility of our approach, we investigate the most widely used technique for defending against adversarial attacks, so-called adversarial training. The defense method requires including adversarial perturbations in the training process by either augmenting the dataset with pre-computed examples (Ilyas et al., 2019) or by incorporating an attack model into the training procedure (Goodfellow et al., 2014; Madry et al., 2018). It has been recently demonstrated that adversarial training performs well on the CIFAR-10 machine learning dataset when enough resources are dedicated to hyperparameter tuning (Gowal et al., 2020), although the performance is still far from that on unperturbed images. We provide a new perspective on this defense by observing that it increases the distance to the decision boundary in the entire adversarial subspace, even though the method utilizes only a single adversarial perturbation per input sample. We additionally find that adversarial training increases the number of alternative label regions in adversarial subspaces, and changes the relative distribution of nearby alternate classes. Herein, we will focus on defining the method and summarizing the subspace with a continued application of understanding adversarial training.

## 2 METHODS

In the following we will outline the novel methods employed in this study. Namely, the method for finding adversarial subspaces is first described, and then we provide an overview of how we measure decision boundary curvature (with a rigorous discussion given in Appendix A). Our primary investigation was conducted using the CIFAR-10 dataset (Krizhevsky &

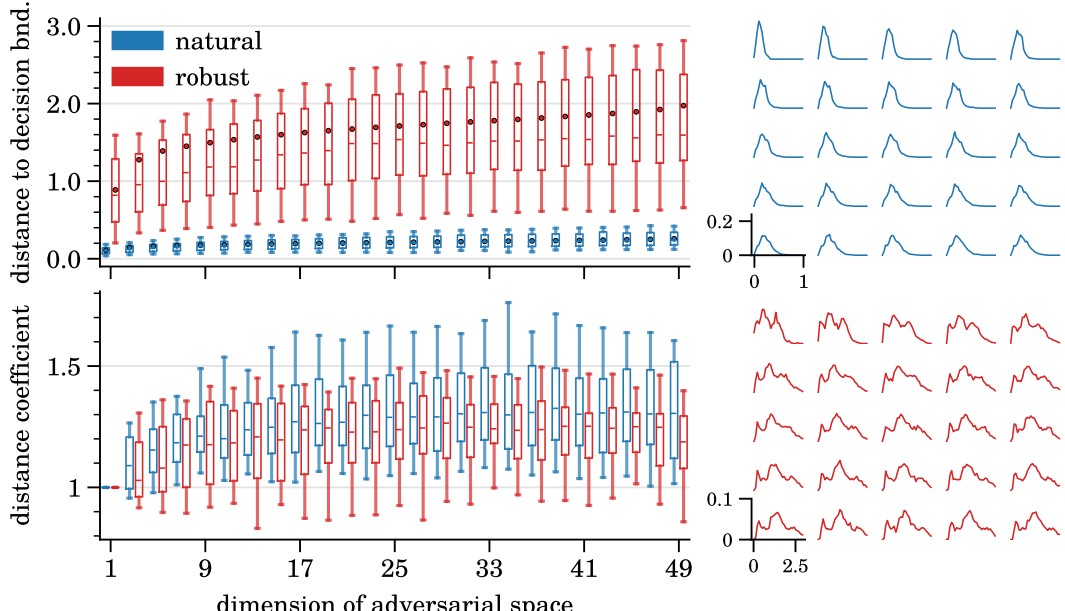

Figure 2: **Distances to the decision boundary.** The distance to the decision boundary for random perturbations within an adversarial subspace grows sub-linearly with increased dimensionality. (Upper left) The dots indicate the adversarial perturbation length for that dimension (i.e. along the perturbation axis), averaged across images. The box plots show the distance to the decision boundary for uniformly-sampled directions within the n-dimensional adversarial subspace, where n is increasing along the horizontal axis. The box indicates the interquartile range, the whiskers indicate the 10% and 90% data percentiles, and the solid horizontal lines indicate the mean. (Lower left) Same as the above, except now the vertical axis indicates the ratio of the largest distance to the decision boundary divided by the length of the given subspace dimension,. That this is small indicates that we rarely sampled a boundary point more than 50% farther away than the largest adversarial dimension. (Right) Distribution of boundary distances as one increases subspace dimensionality, i.e the probability (vertical axis) of a sampled direction having a specific distance (horizontal axis). Observe that the mean shifts to the right, and the variance increases proportionately, which suggests that the distance smoothly increases from one dimension to the next.

Hinton, 2009) and compared two identical ResNet-50 ANNs (He et al., 2016) trained with and without adversarial examples. The networks were modified and retrained using $C^1$ activation functions for the curvature analysis (ELU with $\alpha = 1$). Further dataset and model details can be found in Appendix B. In Appendix C, we provide additional results on the CIFAR-10 dataset, including a comparison with the WideResNet-70 architecture with Swish activations from Gowal et al. (2020). In Appendix D we provide results on the MNIST dataset, which demonstrates largely consistent trends across datasets. In Appendix E we include control tests to establish the consistency of our result for different random seeds. All measured distances are with respect to the $\ell_2$-norm.

## 2.1 FINDING ADVERSARIAL SUBSPACES

We aim to find the set of orthonormal vectors that span the subspace where an input sample has minimal distance to the decision boundary. In order to achieve this, we follow a greedy optimization approach to span an adversarial subspace up to a desired number of dimensions. More precisely, consider some neural network, $f$, that receives an $n$-pixel input image, $x \in \mathbf{R}^n$, and produces an output vector of probabilities associated with $l$ label categories, $\hat{y} \in \mathbf{R}^l$. For a given input, we wish to find an ordered set of orthogonal vectors, i.e. $\Delta = [\delta_1, \delta_2, \ldots, \delta_m], m \leq n$, such that $\langle \delta_k, \delta_m \rangle = 0, \quad \forall k < m$. Thus for each vector, we want to solve an optimization objective that minimizes $||\delta_m||_p$, where $||\delta||_p = (\sum_i |\delta|^p)^{\frac{1}{p}}$ is the $l_p$ norm. We expect the optimizer to explicitly and jointly consider the boundedness of the input

domain, force the perturbations to be orthogonal, and produce perturbations that result in new (incorrect) classifier outputs. Such an objective is intractable to solve for deep neural networks, but is commonly approximated by the untargeted adversarial attack objective[1], which modifies the conditional to be subject to $\max_{j \neq i} f_j(x + \delta) - f_i(x + \delta) > 0$, where $i$ is the correct class index. The length of these vectors is nearly synonymous with distances to the decision boundary, where the actual decision boundary would be an infinitesimal step towards $x$ from $x + \delta$. We must further relax the objective to a differentiable approximation for finding a gradient based solution. While there are many such approximations, we opted for a modification of that which was used by Szegedy et al. (2014) to find singular adversarial directions, which is to minimize the inverse of the cross entropy loss (the training objective of the model), weighed with a distance penalization term. Specifically, $\ell \stackrel{\text{def}}{=} \frac{1}{L_{CE}(x+\delta)} + \kappa \cdot ||\delta||_p$, with the cross entropy loss defined as $L_{CE}(x) \stackrel{\text{def}}{=} -\log \frac{\exp f_i(x)}{\exp \sum_j f_j(x)}$. Using this loss term, we can now define our objective function as implemented:

$$
\begin{aligned}
&\text{minimize} \quad && \frac{1}{L_{CE}(x + \delta_m)} + \kappa \cdot ||\delta_m||_p \\
&\text{subject to} \quad && \max_{j \neq i} f_j(x + \delta_m) - f_i(x + \delta_m) > 0 \\
& && x + \delta_m \in [0, 1]^n \\
& && \langle \delta_k, \delta_m \rangle = 0, \quad \forall k < m.
\end{aligned}
\tag{1}
$$

Intuitively, this maximizes the model loss with respect to the true class label while keeping the perturbation as small as possible. The first constraint in Equation 1 is to ensure that the image is adversarial[2], the second keeps the perturbed images within the allowable pixel bounds, and the last one enforces orthogonality of all found adversarial subspace dimensions. We found the optimal value of constant $\kappa$ in equation (1) with a binary search performed during run time. The optimization problem was solved using the greedy interior-point optimizer implemented by Ipopt (Wächter & Biegler, 2006).

Once the adversarial vectors are found, we conducted experiments to measure the distance to the decision boundary in the space spanned by the vectors (i.e. within the adversarial subspace). The experiment was repeated on 100 test images that were chosen to have equal label representation (10 images per label) and to be correctly classified by both networks. For each test image we found 50 orthogonal adversarial vectors, which allowed us to define a sequentially growing list of adversarial subspaces with dimensionality increasing from 1 to 50. For a given image and subspace we sampled 100 directions via a linear combination of normally distributed variables, which has been previously proven to provide an even sampling (Muller, 1959). Finally, for each random direction we performed a binary search to find the decision boundary.

## 2.2 Measuring decision boundary curvature

As before, let $f_i, f_j : \mathbb{R}^m \to \mathbb{R}$ be the functions corresponding to $i$ and $j$-th logits of a neural network, $f$. Let $F_{ij} \stackrel{\text{def}}{=} f_i - f_j$. Then the $(i, j)$-*decision boundary* is the set

$$
B_{ij} \stackrel{\text{def}}{=} \{x \in \mathbb{R}^n \mid F_{ij}(x) = 0\} = \{x \in \mathbb{R}^n \mid f_i(x) = f_j(x)\}.
\tag{2}
$$

The set $B_{ij}$ is defined independently of the sign of $F_{ij}$ (that is, $B_{ij} = B_{ji}$), although doing so will produce a change of sign in our calculations. Therefore we will continue to follow the above convention by identifying, at a point $x$, $i$ with the correct label associated to $x$, and $j$

---

[1]Many works use "untargeted" adversarial attacks to refer to attacks that are actually targeted to the second-most-likely output class (for example, Carlini & Wagner, 2017; Kurakin et al., 2017; He et al., 2018; Chen et al., 2018). While these may be practically equivalent for a single perturbation, they are not when finding multiple orthogonal directions. Therefore, we will use a more general definition of untargeted attack, following (Tabacof & Valle, 2016; Finlay et al., 2019).

[2]Since we are not targeting any specific class, the first constraint avoids a completely uniform output distribution (i.e. no class wins)

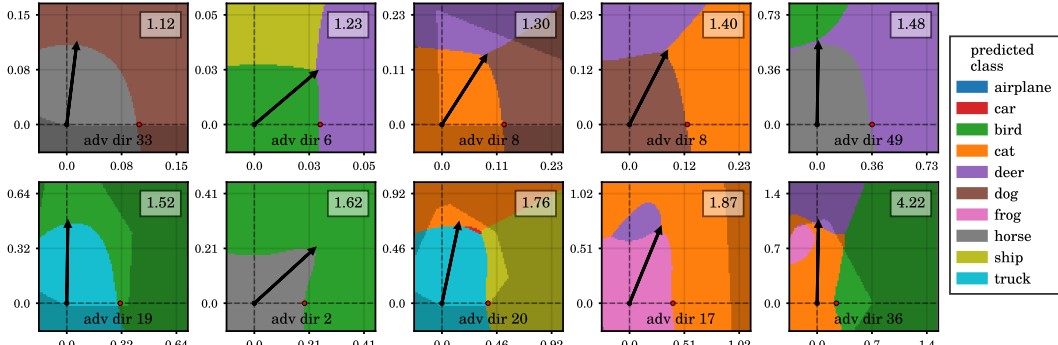

Figure 3: **Boundary visualizations.** Each subplot visualizes a two dimensional cross-section of the decision space for a naturally trained network. Each cross-section is spanned by the vectors corresponding to the largest distance coefficient found for a given input image (visualized as black arrows) and the according adversarial perturbation of the largest dimension of the subspace from which the distance coefficient was retrieved. The red dots indicate the position of the described adversarial perturbation. The number in the top right corner depicts the distance coefficient of the cross-section (as reported in Figure 2, bottom left panel). Areas of the cross-section that lie outside the image bounds are shaded.

with the incorrect label. This convention (which agrees with Moosavi-Dezfooli et al., 2018), ensures that positive (resp. negative) curvature of $B_{ij}$ near $x$ corresponds to $B_{ij}$ bowing inward towards (resp. away from) $x$ (see Figure 1 for an example of positive curvature). If the curvature along $B_{ij}$ is zero near $x$ then $B_{ij}$ is affine. The complete details of how our calculations were derived, and numerical considerations therein, are relegated to Appendix A, however we give a brief intuitive account below.

In our curvature analysis we use a method known as principal curvature decomposition to understand the degree of linearity of the decision boundary. Since the decomposition is performed on a level-set of the decision space, which in general is of codimension 1, it yields $n-1$ scalars and vectors, called the *principal curvatures* and *principle curvature directions*, respectively. The curvature directions point in the directions of maximal and minimal surface curvature, while the principal curvatures themselves indicate the signed magnitude of the curvature. Similar to principal component analysis, which organizes the space as an orthogonal basis of directions of maximal variance, principal curvature analysis organizes the (tangential) space of the decision boundary into orthogonal directions of maximum curvature. The principle curvatures and associated directions are the eigenvectors and eigenvalues of a linear operator $s : \mathbb{R}^m \to \mathbb{R}^m$ called the *shape operator*.

In order to study the curvature over an adversarial (or random) subspace we introduce the *subspace pullback* of $s$

$$\left(P^{\mathsf{T}}\right)^* s \stackrel{\text{def}}{=} P s P^{\mathsf{T}}, \tag{3}$$

where $P \in \mathbb{R}^{m \times n}$ is an orthogonal projection, so that the image of $P^{\mathsf{T}}$ spans the adversarial subspace of $\mathbb{R}^m$ in question. This definition of $\left(P^{\mathsf{T}}\right)^* s$ essentially restricts the shape operator to a linear subspace of the tangential space and is motivated by the principles of differential geometry, details of which can be found in Remark 1 and the surrounding material in Appendix A. The pullback shape operator allows us to compare the amount of curvature in the adversarial subspace to that of the entire response space.

## 3 RESULTS

By definition, the distance to the decision boundary increases for each additional dimension of an adversarial subspace. The circles in the top left panel of Figure 2 indicate these distances, averaged across the test images, which are found by greedily minimizing Equation 1. The distance to the decision boundary grows sub-linearly as one increases the dimensionality of the subspace, with the fastest growth occurring in the first couple of dimensions. The leftmost box plots of the top left panel in figure 2 as well as the Appendix Table 2 confirms

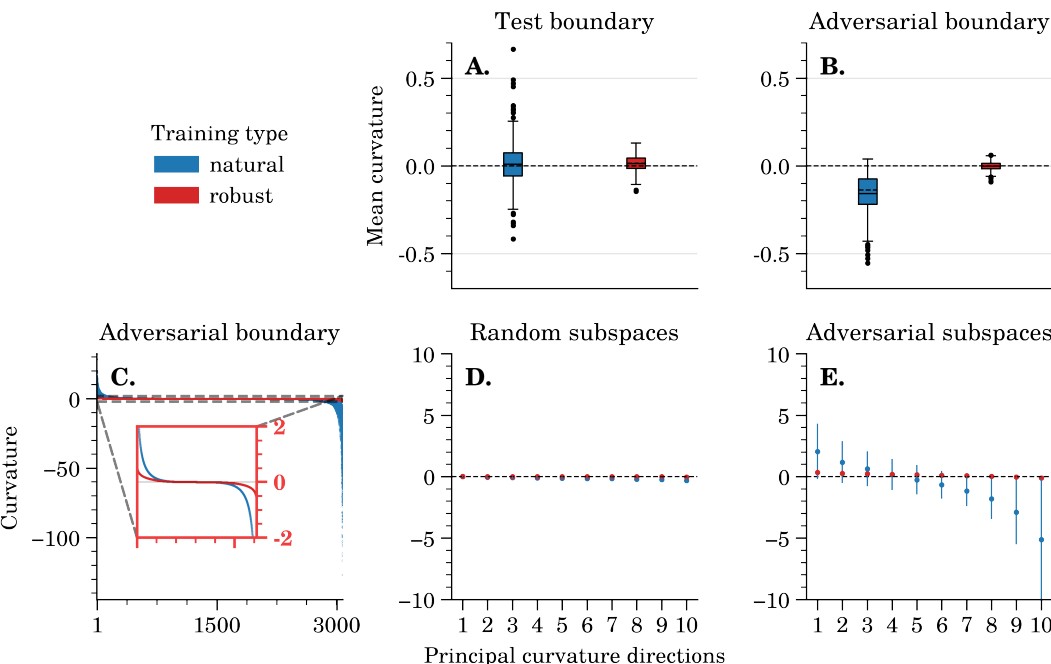

Figure 4: **Decision boundary curvature.** All plots include data for 50 correctly classified test images with evenly distributed labels and the first 10 adversarial subspace dimensions. (A,B) The mean curvatures per boundary point are smaller when the boundary is separating two test images ("test boundary") than when it is separating a test image and an adversarial image ("adversarial boundary"). (C) Scatter plot of the curvature profiles for all images and directions with the adversarial boundary condition. The red inlay shows the full dimensionality (i.e. same horizontal axis as the parent plot), but zoomed in (i.e. different vertical axis limits) to emphasize that while both models have positive and negative curvature, the adversarially trained boundary is more flat (i.e. linear). The data includes all adversarial subspace boundary points per test image. (D,E) Curvature of adversarial point boundaries when restricted to random or adversarial subspaces. Curvature in the adversarial subspace is more extreme than when in an equal sized random subspace. The points indicate the mean across test images and the error lines indicate the standard deviation.

that the mean initial perturbation distances are comparable to more traditional adversarial attacks (Madry et al., 2018) and decision boundary distance estimates (Ding et al., 2020). While this gives an idea of the lower-bound of distances to the decision boundary, it does not provide information about the behavior of the boundary elsewhere in the subspace. To better understand this, we uniformly sampled 100 random directions *within a given subspace* and performed a binary search to find the decision boundary along those directions. We only report distances for decision boundary points that are within the $\ell_{\text{inf}}$ allowable pixel box, which was never fewer than 40% of the subspace samples (see Appendix C Figure 7). The box plots in the top left panel of Figure 2 indicate the distribution of decision boundary distances across the sampled directions and test images. There is no consistent $\ell_2$-length of an adversarial attack that makes an perturbation visible to a human observer, although in general the perturbations were visible after around dimension 20 for the robust network and beyond dimension 50 for the natural network (see Appendix C Figure 6 for examples). The bottom left panel of Figure 2 plots a distance coefficient that is defined as the maximum distance to the decision boundary within an adversarial subspace divided by the longest vector defining the subspace (which is always associated with the highest dimension number). For example, a value of 2 would indicate that the largest sampled distance is twice as far away as the largest found adversarial perturbation. The adversarial trained network consistently produces smaller coefficients, indicating that the boundary size is less variable between the subspace axes. Finally, a complete description of the boundary distances is conveyed as histograms in the rightmost panels, with dimensionality increasing from left to right, then top to bottom. Together, this provides further evidence that the decision boundary grows

consistently, without large variations in distance between the found adversarial subspace axes.

In Figure 3 we visualize two-dimensional cross-sections of the decision surface. The horizontal axis of each cross-section corresponds to an adversarial subspace perturbation, the black arrow indicates the farthest decision boundary point, and the vertical axis is found by orthogonalizing the direction of the farthest decision boundary point with respect to the adversarial perturbation using one step of the Gram-Schmidt process. The cross-sections are ordered according to the distance coefficient, which is displayed in the upper-right corner. We emphasize that because these are cross-sections, they avoid the shortcomings of typical low-dimensional decision space projections noted in the introduction. Among the visualizations one can see examples of when the boundary increases smoothly from one axis to the next, and also the more rare (as indicated by Figure 2) cases when the boundary protrudes quite far from the origin when between the axes. From this analysis we conclude that adversarial examples are not isolated or due to sharp protrusions into an otherwise homogeneously labeled region. Rather, there exists a smooth increase in distance to the decision boundary from the extreme case of an initially found adversarial example to semantically differentiated images.

In Figure 4 we measure boundary curvature for several experimental conditions that vary the type of training, the location of the boundary, and the subspace defining vectors. Panels A and B report the mean curvature (i.e. the mean of the eigenvalues of the shape operator) per point on the decision boundary. Both model types have larger boundary curvature for boundary points separating clean images from adversarial images (adversarial boundary; Panel B) when compared to boundary points separating unperturbed test set examples (test boundary; Panel A). We additionally found that adversarial training tends to linearize the decision boundary, both between correctly labeled test images and near adversarial examples. Panel C is a scatter plot of the principal curvatures (i.e. eigenvalues of the shape operator) in the high-dimensional input space for all of the adversarial boundary points. We consistently find both positive and negative curvature at each boundary point and for both training types. Using the pullback defined in Equation 3, we can measure the subspace principal curvatures for random (Panel D) and adversarial (Panel E) subspaces. The curvature within adversarial subspaces is notably larger than in random subspaces. The difference in curvature values between naturally and adversarially trained networks is less pronounced when limited to random subspaces – suggesting that much of the linearization occurring during adversarial training is along directions aligned with adversarial examples. The found boundary points can be the closest to the clean input even when there is positive curvature, as long as the radius of curvature is greater than the distance to the input. However, for naturally trained networks, adversarial boundary points tend to have more negative mean curvature, both in the full input space (Panel B) and in the adversarial subspace (Panel E).

Table 1: Entropy of unique proximal adversarial labels per class, averaged across a balanced sampling of correctly labeled test images.

| Label | Natural | Robust |
|---------|---------|--------|
| airplane | 1.70 | **1.81** |
| car | 1.01 | **1.26** |
| bird | **1.72** | 1.59 |
| cat | 1.56 | **1.97** |
| deer | 1.41 | **1.98** |
| dog | 1.51 | **1.82** |
| frog | 1.12 | **1.51** |
| horse | 1.63 | **1.78** |
| ship | **1.74** | 1.39 |
| truck | 1.79 | **1.88** |

In addition to the earlier comparisons between natural and adversarial network training, we measured the diversity of adversarial classes near test samples. The leftmost subplot of Figure 5 gives the number of unique adversarial classes found in the first 50 dimensions of the adversarial subspaces, averaged across correctly labeled test images. The other plots show the distribution of adversarial classes present in the subspaces with respect to the origin label. These plots suggest that adversarial training results in an increased diversity of alternate classes near any given test sample, which we further quantify by measuring the discrete entropy of each distribution in Table 1. Thus, in addition to (or more probably as a consequence of) increasing the distance to the boundary, adversarial training further modifies the density of alternative class labels near test images. While this could be considered a

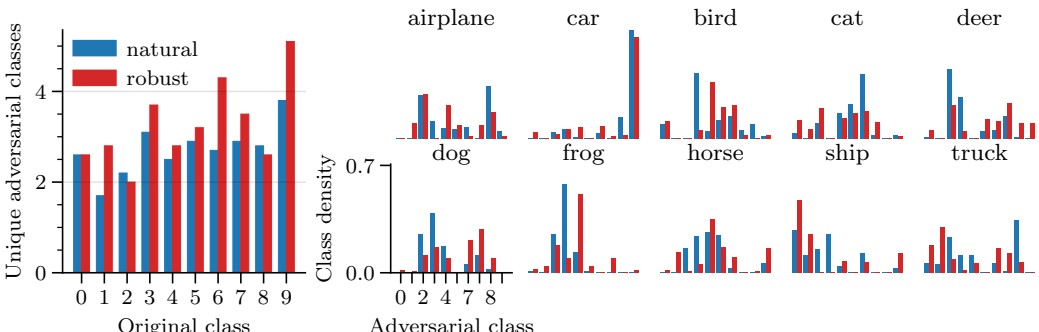

Figure 5: **Adversarial Class Composition.** (Left) Within the adversarial subspaces of randomly chosen test images we quantified how many unique adversarial classes exist. In most cases adversarial training increases the diversity of alternate classes close to any test image. (Right) Each plot shows the distribution of nearby alternate classes for each origin class with natural and adversarial training. Adversarial training results in a more even distribution of nearby classes for all categories except bird and ship.

desired effect, as it indicates that the robust network utilizes more of the high-dimensional space, it also leads to behavior that is less matched to human behavior (something that is often sought along with adversarial robustness). Indeed, while some categories, like "car" and "truck" are intuitively adjacent, the distribution of classes generally failed to match any clear intuitions. For example, observe the lack of symmetry between cars and trucks, or that adversarial training switches the "frog" category's most frequent alternate class from "bird" to "cat".

## 4  RELATED WORK

**Related adversarial subspace studies:** The work of Tramèr et al. (2017) is most similar to our adversarial decomposition, in that they reported the dimensionality of a subspace in the proximity of a given adversarial perturbation, although they do not comment on the class, number, or proximity of nearby orthogonal adversarial examples. Work from Ma et al. (2018) measured the local intrinsic dimensionality (LID) of adversarial subspaces, which indirectly estimates adversarial subspace dimensionality by measuring the distance from an adversarial example to a batch of nearest unperturbed dataset examples. This idea has been extended to algorithms for detecting adversarial examples (Ma et al., 2018; Mao et al., 2020), although Athalye et al. (2018) generated adversarial examples with indistinguishable LID scores from natural examples. In terms of understanding ANN classifiers, such an approach leaves open the questions of how proximal the decision boundary is from input samples in greater than one dimension or the actual dimensionality of adversarial subspaces. Our approach is the first to examine the decision boundary where it is nearest to test samples by defining adversarial subspaces using multiple orthogonal adversarial directions. Our emphasis on the subspace where the decision boundary is closest to test images is also unique when compared to work that characterizes the decision boundary in random directions, which will be considerably farther away from the origin (e.g. Fawzi et al., 2016; Tabacof & Valle, 2016; Liu et al., 2017; He et al., 2018). Finally, most of these studies perform a clipping operation *after* defining or finding the desired directions, which results in a non-orthogonal set (see Appendix F for a more detailed discussion). We overcome this problem by including the valid pixel range as inequality constraints in the optimization process. Our work is complementary to previous studies that investigate the relationship between adversarial robustness and *input* dimensionality (Amsaleg et al., 2017; Simon-Gabriel et al., 2019), since the comparative trends observed between robust and natural networks are relatively unchanged with small (MNIST) or large (CIFAR) dimensional inputs.

**Neuron response curvature:** The use of response curvature to understand nonlinear functions has a long history in neuroscience (for a review, see Gollisch & Herz, 2012) as well as models of neural computation (e.g. Zetzsche & Krieger, 2001; Rust et al., 2005; Golden et al., 2016; Cohen et al., 2020). The principal curvature decomposition was previously employed in this context by Golden et al. (2019), who used it to understand invariance

and selectivity of individual ANN neurons, and by Paiton et al. (2020), who first found a relationship between individual neuron response curvature and adversarial robustness. Our work is distinct from the aforementioned studies in that we investigate the curvature of the decision boundary, as opposed to that of individual neurons. However, the relationship between decision boundary curvature and adversarial robustness has been previously studied by Moosavi-Dezfooli and colleagues, who provided a number of compelling results identifying "an increasing [adversarial] vulnerability with respect to the curvature of the decision boundary" (Moosavi-Dezfooli et al., 2018). As we discussed in the previous section, our analysis provides support for hypotheses that go beyond the correlation between decision boundary curvature and network robustness, in that we observe different boundary curvature in adversarial subspaces than random subspaces or the full input space. We additionally note that while previous work has utilized Riemannian curvature analysis to understand neural networks (viz. Fawzi et al., 2016; Poole et al., 2016; Moosavi-Dezfooli et al., 2018; 2019; Golden et al., 2019), each study measures different quantities without complete details of the derived formulas. We demonstrate in Appendix G that these formulas do not find the correct level-set curvature of simple manifolds. This together with our rigorous derivation and discussion in Appendix A constitutes, to the best of our knowledge, the first principled and analytically supported measurement of decision boundary curvature of trained nonlinear neural networks.

## 5 Conclusion

There is an ongoing debate on the cause, prevalence, and uniformity of adversarial examples. Several studies have supported a hypothesis that most adversarial examples inhabit relatively dense regions of the input space (Goodfellow et al., 2014; Tabacof & Valle, 2016; Moosavi-Dezfooli et al., 2017a; He et al., 2018). However, other works have suggested that adversarial examples live in isolated pockets or thin "spikes" of incorrect classification regions (Szegedy et al., 2014; Yu et al., 2019b). Here we provide quantitative evaluation supporting the hypothesis that decision boundaries are close to input samples in a large subspace. Additional work has suggested that it is the linearity of deep neural networks that leads to adversarial examples (Goodfellow et al., 2014), and that adversarial training leads to a less linear decision boundary (Madry et al., 2018). However, more recent work has proposed the alternative: that increasing decision boundary linearity causes increased robustness (Li et al., 2019b; Moosavi-Dezfooli et al., 2019; Terjék, 2020; Sarkar & Iyengar, 2020). Our curvature analysis confirms that adversarial training linearizes the boundary, which is also congruent with hypotheses about the robustness benefits of smoothing the decision surface (Zhang et al., 2019; Wu et al., 2020). We further compare the boundary curvature in adversarial subspaces and random subspaces to show an increase in curvature in adversarial subspaces for both training methods. Since we do not report the causal relationship between adversarial directions and boundary curvature, our results have no impact on hypotheses for adversarial susceptibility due to overfitting (Tanay & Griffin, 2016) or the perceptual quality of adversarial features (Ilyas et al., 2019), although we see this as an interesting future direction of study.

Adversarial subspaces allow for a quantitative description of the nonlinear geometry and class composition of the network decision boundary in the subspace spanned by its most susceptible perturbation directions. They also assist in explaining the prevalence of adversarial examples as well as the effects of adversarial training. Understanding this space better can improve neural network performance with semi-supervised relabeling (Benato et al., 2018), architecture design (Rauber et al., 2018), and adversarial robustness (Tramèr et al., 2017; Moosavi-Dezfooli et al., 2019). Importantly, we also provide a quantitative and falsifiable interpretability approach for understanding ANN decisions (Ribeiro et al., 2016; Vlassopoulos et al., 2020), which is a necessary prerequisite for the technology to be applied to many industries (Doshi-Velez & Kim, 2017; Leavitt & Morcos, 2020). We have identified an interesting avenue for future work in performing a large-scale analysis of the adversarial subspaces of various neural network architectures, as well as estimating decision boundary geometry of non-smooth (e.g. ReLU or stochastic, Li et al., 2019a; Dapello et al., 2020) neural networks (which requires adding a costly Hessian approximation step (Moosavi-Dezfooli et al., 2019)).

AUTHOR CONTRIBUTIONS

ACKNOWLEDGMENTS

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

## A  BOUNDARY CURVATURE ANALYSIS

The following definitions are common. Our notation follows Lee (2013; 2018) for differential geometry, and Magnus & Neudecker (1999) for linear algebra.

### A.1  INTRODUCTION

In order to derive our main curvature expressions for a graph submanifold, we first introduce some notation and recall some concepts.

**Tangent Spaces**  If $M$ is an $n$-dimensional smooth manifold then the *tangent space at $p \in M$* is a vector space $T_p M$ of derivations at $p$, which is isometric to $\mathbb{R}^n$. In particular, $T_p M$ is the set of directional derivative operators acting on the smooth real functions $f : M \to \mathbb{R}$.

The disjoint union of all of the tangent spaces is the *tangent bundle* $TM \stackrel{\text{def}}{=} \{\{p\} \times T_p M \mid p \in M\}$. If we equip $M$ with a smooth, bilinear, positive definite, symmetric form $g : TM \times TM \to \mathbb{R}$ then $(M, g)$ is said to be a *Riemannian manifold* and $g$ the *Riemannian metric tensor.*

When it is unambiguous, denote by $\Gamma(E)$ the smooth mappings $M \to E$ (for some topological space $E$). Most often we will refer to $\mathfrak{X}(M) \stackrel{\text{def}}{=} \Gamma(TM)$, and represent elements of $\mathfrak{X}(M)$ with the letters $X, Y, Z$.

**Differentials**  If $M$ and $N$ are manifolds and $F : M \to N$ is differentiable, then the *differential of $F$ at $p \in M$* is the linear mapping denoted $dF_p : T_p M \to T_p N$, where the action on tangent vectors is defined by

$$\forall_{p \in M} \forall_{v \in T_p M} \forall_{f \in C^1(N)} : \ dF_p(v) f = v(f \circ F)(p).$$

That is, the rate of change of the function $f \circ F$ in the direction $v \in T_p M$ at the point $p$. As is common practice, we often omit the $p$ subscripts for clarity.

The linear operator $df$ is a linear functional when $N$ in the above definition is $\mathbb{R}$, and at each $p \in M$ there exists a vector $\operatorname{grad} f_p \in T_p$. So that

$$\forall_{p \in M} \forall_{v \in T_p M} : \ df_p(v) = g(\operatorname{grad} f_p, v),$$

and $\operatorname{grad} f \in \mathfrak{X}(M)$ is the associated vector field.

**Connections**  A *connection* is an operator $\nabla : \mathfrak{X}(M) \times \mathfrak{X}(M) \to \mathfrak{X}(M)$, linear in its first argument, and satisfying a product rule in its second. Surprisingly, a connection uniquely defines a connection in each tensor bundle,[3] (which we overload with $\nabla$) (Lee, 2018,

---

[3]Including the $(0,0)$-tensor bundle, that is, the smooth real functions on $M$.

Prop. 4.15). Importantly for us, we have $\nabla_X f = Xf$ for all $X \in \mathfrak{X}(M)$ (for any connection).[4] A Riemannian manifold always posesses a *Levi–Cevita connection* (Lee, 2018, §5), denoted by $\nabla$, which is the unique connection satisfying

$$\forall_{X,Y,Z \in \mathfrak{X}(M)}: \ \nabla_X g(Y,Z) = g(\nabla_X Y, Z) + g(Y, \nabla_X Z).$$

To $f \in \mathrm{C}^2(M)$ we associate the Hessian operator $\nabla^2 f$, which is a symmetric $(0,2)$-tensor field (Lee, 2018, Ex. 4.22) with $\nabla^2 f(X,Y) \stackrel{\text{def}}{=} \nabla_X \, \mathrm{d}f(Y)$ for $X,Y \in \mathfrak{X}(M)$. We have (via Lee, 2018, Prop. 4.21)

$$\forall_{X,Y \in \mathfrak{X}(M)}: \ \nabla_X \, \mathrm{d}f(Y) = \nabla_X(\nabla_Y f) - \nabla_{(\nabla_X Y)} f = X(Yf) - (\nabla_X Y)f. \tag{4}$$

Using the compatibility of $\nabla$ with the metric we arrive at the following expression for the Hessian, which will be useful later

$$\begin{aligned}
\nabla^2 f(X,Y) &\stackrel{(4)}{=} X(Yf) - (\nabla_X Y)f \\
&= \underbrace{X g(\operatorname{grad} f, Y)}_{\nabla_X g(\operatorname{grad} f, Y)} - g(\operatorname{grad} f, \nabla_X Y) \\
&= g(\nabla_X \operatorname{grad} f, Y) + g(\operatorname{grad} f, \nabla_X Y) - g(\operatorname{grad} f, \nabla_X Y) \\
&= g(\nabla_X \operatorname{grad} f, Y). \tag{5}
\end{aligned}$$

### A.1.1 SUBMANIFOLDS

If $(M, g)$, $(\tilde{M}, \tilde{g})$ are two Riemanian manifolds with $M \subseteq \tilde{M}$, then an injection $\iota : M \hookrightarrow \tilde{M}$ is called an *inclusion mapping.* If the inclusion is a smooth isometry, that is

$$\forall_{u,v \in \mathrm{T}M}: \ g(u,v) = \tilde{g}(\mathrm{d}\iota(u), \mathrm{d}\iota(v)),$$

equivalently, $g = \iota^* \tilde{g}$, then $M$ is called a *Riemanian submanifold of $\tilde{M}$.*

Using the inclusion map differential, we embed $\mathrm{T}M$ in $\mathrm{T}\tilde{M}$. If $M$, $\tilde{M}$ are of dimensions $m < n$, then $\mathrm{T}_p M$ occupies an $m$ dimensional subspace of $\mathrm{T}_p \tilde{M}$ for each $p \in M$. The *normal bundle* is formed from the orthogonal compliment of $\mathrm{T}_p M$ at each $p \in M$, and is defined in a similar way to the tangent bundle, denoted $\mathrm{N}M$ and having a dimension of $n - m$. For vectors $v \in \mathrm{T}\tilde{M}$ we have the decomposition $v = v^\top + v^\perp$ with $v^\top \in \mathrm{T}M$ and $v^\perp \in \mathrm{N}M$.

Some care must be taken when taking covariant derivatives in $\mathrm{T}M$ with the ambient connection, formally

$$\forall_{X,Y \in \mathrm{T}M}: \ \tilde{\nabla}_X Y \stackrel{\text{def}}{=} \tilde{\nabla}_{\overline{\mathrm{d}\iota(X)}} \overline{\mathrm{d}\iota(Y)},$$

where the overlines denote arbitrary smooth extensions of $\mathrm{d}\iota(X)$ and $\mathrm{d}\iota(Y)$ to open neighbourhoods of $M$ in $\tilde{M}$.

**Second Fundamental Form** Let $M \subseteq \tilde{M}$ be an embedded sumbanifold. The *second fundamental form* is the (bilinear) mapping $\mathrm{II} : \mathfrak{X}(M) \times \mathfrak{X}(M) \to \Gamma(\mathrm{N}M)$ with $\mathrm{II}(X,Y) \stackrel{\text{def}}{=} (\tilde{\nabla}_X Y)^\perp$. If we equip $M$ with a normal field $N \in \Gamma(\mathrm{N}M)$, then the *scalar second fundamental form* is

$$\mathrm{II}_N(X,Y) \stackrel{\text{def}}{=} \tilde{g}(N, \mathrm{II}(X,Y)).$$

For the remainder we assume $N$ is fixed. The *shape operator* $s : \mathfrak{X}(M) \to \mathfrak{X}(M)$ is the endomorphism obtained from $\mathrm{II}_N$ that satisfifes

$$\forall_{X,Y \in \mathfrak{X}(M)}: \ \mathrm{II}_N(X,Y) = g(sX,Y), \tag{6}$$

that is, by raising an index.

---

[4]That is, the action of a vector field through the connection agrees with the ordinary directional derivative.

**Curvature**  Let $s$ be the shape operator associated to a normal field on an embedded submanifold $M$ of dimension $n$. Then at $p \in M$, the (real) eigenvalues of $s$ are called *principal curvatures* and the eigenvectors are called *principal directions* at $p$. The *Gaussian curvature* is the product of the principal curvatures, $\det(s)$, and the *mean curvature* is their average, $\frac{1}{n} \operatorname{tr}(s)$.

Assume $(M, g)$ and $(\tilde{M}, \tilde{g})$ are Riemannian manifolds with an inclusion mapping $\iota : M \hookrightarrow \tilde{M}$ and $M$ has the submanifold structure. Suppose $\tilde{M}$ posesses a shape operator (or any other linear endomorphism) $\tilde{s} : T\tilde{M} \to T\tilde{M}$. The submanifold structure then gives us an operator $\iota^* \tilde{s} : TM \to TM$, the pullback of $s$ by $\iota$, via

$$\forall_{X \in TM} : \ \iota^* s X = \mathrm{d}\iota^{-1} s \, \mathrm{d}\iota(X).$$

*Remark* 1 (Euclidean subspace). We consider the special case of the Riemannian manifold $\mathbb{R}^m$. Suppose $P : \mathbb{R}^n \to \mathbb{R}^m$ is orthogonal ($n \leq m$) and let $s : \mathbb{R}^m \to \mathbb{R}^m$ be a linear operator. Then $P^\mathsf{T} : \mathbb{R}^n \hookrightarrow \mathbb{R}^m$ is an inclusion map and the *subspace pullback* of $s$ to $\mathbb{R}^n$ is

$$\left(P^\mathsf{T}\right)^* s = P s P^\mathsf{T}.$$

We refer to the curvature quantities and directions of $\left(P^\mathsf{T}\right)^* s$ with the suspace prefix. That is, *subspace principal curvature*, *subspace Gaussian curvature*, and so on.

## A.2  Hypersurfaces

Let $(M, g)$ be an embedded submanifold of $(\tilde{M}, \tilde{g})$. Suppose $F : U \subseteq \tilde{M} \to \mathbb{R}$ is a *local defining function for $M$*, that is, $M \cap U = F^{-1}(\{0\})$ for an open $U \subseteq M$. Choose arbitrary $X, Y \in \mathfrak{X}(M)$. From the connection product rule (Lee, 2018, p. 89)

$$\tilde{\nabla}_X \left( \tfrac{1}{|\operatorname{grad} F|} \cdot \operatorname{grad} F \right) = \frac{1}{|\operatorname{grad} F|} \tilde{\nabla}_X \operatorname{grad} F + \left( X \tfrac{1}{|\operatorname{grad} F|} \right) \cdot \operatorname{grad} F.$$

Whence

$$\tilde{g}\big(\tilde{\nabla}_X N, Y\big) = \frac{\tilde{g}\big(\tilde{\nabla}_X \operatorname{grad} F, Y\big)}{|\operatorname{grad} F|} + \left( X \frac{1}{|\operatorname{grad} F|} \right) \underbrace{\tilde{g}(\operatorname{grad} F, Y)}_{0}$$

$$= \frac{\tilde{g}\big(\tilde{\nabla}_X \operatorname{grad} F, Y\big)}{|\operatorname{grad} F|}. \tag{7}$$

The zero in the underbrace follows because $\operatorname{grad} F \in NM$ and $Y \in TM$. The Weingarten equation (Lee, 2018, Thm. 8.13(c)) yields $sX = -\tilde{\nabla}_X N$, giving us

$$-\mathrm{II}_N(X, Y) \overset{(6)}{=} -g(sX, Y) = \tilde{g}\big(\tilde{\nabla}_X N, Y\big) \overset{(7)}{=} \frac{\tilde{g}\big(\tilde{\nabla}_X \operatorname{grad} F, Y\big)}{|\operatorname{grad} F|} \overset{(5)}{=} \frac{\tilde{\nabla}^2 F(X, Y)}{|\operatorname{grad} F|}. \tag{8}$$

### A.2.1  The Curvature of a Graph Manifold

The notation and conventions of this section closely follow (Lee, 2018, §8). Let $M$ be the graph of a smooth function $f : \mathbb{R}^n \to \mathbb{R}$, that is, $M = \{(x, f(x)) \mid x \in \mathbb{R}^n\}$. Since $f$ is smooth $M$ is indeed a manifold (Lee, 2013, Ex. 1.30, p. 20). We regard $(M, g)$ as a Riemannian submanifold of $(\mathbb{R}^{n+1}, \langle \cdot, \cdot \rangle)$, with the inclusion map $\iota : M \hookrightarrow \mathbb{R}^{n+1}$.

The function $F : \mathbb{R}^{n+1} \to \mathbb{R}$ is a global defining function for $M$ with $F(x, t) \overset{\text{def}}{=} f(x) - t$. Denote by $\nabla f$ the vector of partial derivatives of $f$, and by $\mathrm{H}_f$ and $\mathrm{H}_F$, the matrices of second order partial derivatives of $F$ and $f$, all with respect to usual coordinate frames, $(x^i)$, on $\mathbb{R}^n$ and $\mathbb{R}^{n+1}$. Then

$$\forall_{X, Y \in T\tilde{M}} : \ \nabla^2 F(X, Y) = \langle \mathrm{H}_F X, Y \rangle.$$

At $p \in M$ the differential $\mathrm{d}\iota_p$ has the Jacobian matrix

$$\big( \mathrm{Id} \quad \nabla f(x^1(p), \dots, x^n(p)) \big)^\mathsf{T}.$$

We commit the mild sin of identifying $\mathrm{d}\iota_p$ with its Jacobian and $X, Y \in \mathrm{T}\tilde{M}$ with their parameterizations under the coordinate frame, so that we may use the notation of matrix multiplication.

In dropping the $p$-subscripts, the left inverse, $(\mathrm{d}\iota)^{-1}$, is has the parameterization

$$\left((\mathrm{d}\iota)^{\mathsf{T}}(\mathrm{d}\iota)\right)^{-1}(\mathrm{d}\iota)^{\mathsf{T}} = \left(\mathrm{Id} + \nabla f \, \nabla f^{\mathsf{T}}\right)^{-1}(\mathrm{d}\iota)^{\mathsf{T}}.$$

Thus

$$(\mathrm{d}\iota)^{\mathsf{T}} \mathrm{H}_F (\mathrm{d}\iota) = \begin{pmatrix} \mathrm{Id} \\ \nabla f^{\mathsf{T}} \end{pmatrix}^{\mathsf{T}} \begin{pmatrix} \mathrm{H}_F & 0 \\ 0 & 0 \end{pmatrix} \begin{pmatrix} \mathrm{Id} \\ \nabla f^{\mathsf{T}} \end{pmatrix} = \mathrm{H}_f,$$

whence

$$\begin{aligned}
(\mathrm{d}\iota)^{-1} \mathrm{H}_F (\mathrm{d}\iota) &= \left(\mathrm{Id} + \nabla f \, \nabla f^{\mathsf{T}}\right)^{-1}(\mathrm{d}\iota)^{\mathsf{T}} \mathrm{H}_F (\mathrm{d}\iota) \\
&= \left(\mathrm{Id} + \nabla f \, \nabla f^{\mathsf{T}}\right)^{-1} \mathrm{H}_f.
\end{aligned} \tag{9}$$

We can now compute the shape operator of $M$ associated to the unit normal field $\operatorname{grad} F / |\operatorname{grad} F|$. Pick arbitrary $X, Y \in \mathrm{T}M$. From (8) there is

$$\begin{aligned}
-\mathrm{II}_N(X, Y) \cdot |\operatorname{grad} F| &\stackrel{(8)}{=} \tilde{g}(\tilde{\nabla}_X \operatorname{grad} F, \mathrm{d}\iota(Y)) \\
&= g\left(\mathrm{d}\iota^{-1} \circ \mathrm{H}_F \circ \mathrm{d}\iota(X), Y\right),
\end{aligned} \tag{10}$$

and, using the bilinearity of of $g$ we have

$$g(sX, Y) \stackrel{(6)}{=} \mathrm{II}_N(X, Y) \stackrel{(10)}{=} -\frac{g(\mathrm{d}\iota^{-1} \mathrm{H}_F \, \mathrm{d}\iota X, Y)}{|\operatorname{grad} F|}.$$

This, together with (9) yields the shape operator:

$$s = -\frac{\left(\mathrm{Id} + \nabla f \, \nabla f^{\mathsf{T}}\right)^{-1} \mathrm{H}_f}{\sqrt{|\nabla f|^2 + 1}}. \tag{11}$$

Whence the Gaussian and mean curvature are the scalar fields

$$-\frac{\det\left(\left(\mathrm{Id} + \nabla f \, \nabla f^{\mathsf{T}}\right)^{-1} \mathrm{H}_f\right)}{\left(|\nabla f|^2 + 1\right)^{\frac{n}{2}}} \quad \text{and} \quad -\frac{\operatorname{tr}\left(\left(\mathrm{Id} + \nabla f \, \nabla f^{\mathsf{T}}\right)^{-1} \mathrm{H}_f\right)}{n\left(|\nabla f|^2 + 1\right)^{\frac{1}{2}}} \tag{12}$$

respectively.

### A.2.2 THE CURVATURE OF A LEVEL-SET MANIFOLD

We now consider the case where $M$ is the level-set of a smooth function $f : \mathbb{R}^n \to \mathbb{R}$ so that for some $c \in \mathbb{R}$ we have $M = \{x \in \mathbb{R}^n \mid f(x) = c\}$. The following theorem can be used to ensure the level-sets of $f$ do indeed posses a manifold structure, which is a simplification of the constant rank level-set theorem (Lee, 2013, Thm. 5.12) for our setting.

**Theorem 1.** *Let $f : \mathbb{R}^n \to \mathbb{R}$ be a smooth function having constant rank $r$. Then the level-sets of $f$ are all smooth manifolds of codimension $r$.*

We use the same construction as Lee (2013, Ex. 1.32, §5) to calculate the Gaussian and mean curvature at a point $(x_0, y_0) \in (\mathbb{R}^{n-1} \times \mathbb{R}) \cap M$.

From the implicit function theorem (Lee, 2013, Thm. C.40) there are open connected neighbourhoods $U_0 \times V_0 \subseteq \mathbb{R}^{n-1} \times \mathbb{R}$ with $(x_0, y_0) \in U_0 \times V_0$ and a smooth function $g : U_0 \to V_0$ so that $(U_0 \times V_0) \cap M$ is the graph of $g$. That is, $(U_0 \times V_0) \cap M = \left\{(x, g(x)) \mid x \in U_0 \subseteq \mathbb{R}^{n-1}\right\}$. Consequentially the gradient of $g$ has the structure

$$\begin{aligned}
\forall_{x \in U_0} : \ f(x, g(x)) = c &\implies \mathrm{D}_1 f(x, g(x)) + \mathrm{D}_2 f(x, g(x)) \nabla g(x) = 0 \\
&\iff \nabla g(x) = -[\mathrm{D}_2 f]^{-1}(x, g(x)) \, \mathrm{D}_1 f(x, g(x)). \tag{13}
\end{aligned}$$

Let $\partial_i \overset{\text{def}}{=} \partial/\partial x^j$ and $\partial_{ij}^2 \overset{\text{def}}{=} \partial^2/\partial x^i\,\partial x^j$. Then for $i,j \in [n-1]$ and $x \in U_0$

$$\partial_i g(x) = -(\partial_n f(x, g(x)))^{-1}\partial_i f(x, g(x)), \tag{14}$$

and

$$
\begin{aligned}
\partial_{ij}^2 g(x) &= \frac{1}{(\partial_n f)^2}\Big[\partial_{nj}^2 f + \partial_{nn}^2 f \partial_j g\Big](\partial_i f) - \frac{1}{\partial_n f}\Big[\partial_{ij}^2 f + \partial_{nj}^2 f \partial_j g\Big]\\
&= -\frac{1}{\partial_n f}\Big[\partial_{nj}^2 f + \partial_{nn}^2 f \partial_j g\Big](\partial_i g) - \frac{1}{\partial_n f}\Big[\partial_{ij}^2 f + \partial_{nj}^2 f \partial_j g\Big]\\
&= -\frac{1}{\partial_n f}\Big[\partial_{nj}^2 f \partial_i g + \partial_{nn}^2 f \partial_i g \partial_j g + \partial_{ij}^2 f + \partial_{nj}^2 f \partial_j g\Big]\\
&= -\frac{1}{\partial_n f}\Big[\partial_{nj}^2 f(\partial_i g + \partial_j g) + \partial_{nn}^2 f \partial_i g \partial_j g + \partial_{ij}^2 f\Big], \tag{15}
\end{aligned}
$$

where every function on the right hand side is evaluated at either the point $x$ or $(x, g(x))$ (depending on the function).[5] If we let $\bar\nabla \overset{\text{def}}{=} (\partial/\partial x^1, \ldots, \partial/\partial x^{n-1})^{\mathsf{T}}$, and $\bar{\mathrm{H}}$ be, similarly, the $(n-1)\times(n-1)$ matrix of cross-partial derivatives, there are the more compact expressions for (14) and (15):

$$\nabla g = -\frac{1}{\partial_n f}\bar\nabla f,$$

and

$$\mathrm{H}_g = -\frac{1}{\partial_n f}\Big[\big(\nabla g + \nabla g^{\mathsf{T}}\big)\operatorname{diag}\big(\bar\nabla \partial_n f\big) + \partial_{nn}^2 f \cdot \big(\nabla g \nabla g^{\mathsf{T}}\big) + \bar{\mathrm{H}}_f\Big],$$

where the sum $\nabla g + \nabla g^{\mathsf{T}}$ is to be interpreted as the matrix of pairwise sums of the elements of $\nabla g$: $\big(\nabla g + \nabla g^{\mathsf{T}}\big)_{ij} \overset{\text{def}}{=} \partial g/\partial x^i + \partial g/\partial x^j$. Then, as in Section A.2.1, $F(x) \overset{\text{def}}{=} g(x^1, \ldots, x^{n-1}) - x^n$ is a local defining function for $M$ on $(U_0 \times V_0) \cap M$. The local graph parameterization provided by $g$ is $\phi : U_0 \to M$ with $\phi(x) \overset{\text{def}}{=} (x, g(x))$, and its differential has the parameterization

$$\mathrm{d}\phi = \begin{pmatrix} \mathrm{Id} & \nabla g \end{pmatrix}^{\mathsf{T}}.$$

Finally, *mutatis mutandis*, the shape operator is given by (11) and the Gaussian and mean curvature can be computed as in (12).

*Remark* 2 (Numerical considerations). In order to compute the level-set curvature we needed to apply the implicit function theorem. However, it may be difficult numerically to stabilize the inverse operation in (13). To remedy this problem, one may apply a change of basis before computing the quantities (14) and (15). Let $B \in \mathbb{R}^{n\times n}$ be an orthogonal change of basis operator. Then we apply all our operations to $B^{\mathsf{T}} M \overset{\text{def}}{=} \big\{B^{\mathsf{T}} x \mid x \in M\big\}$.

Differentiating $\tilde{f} \overset{\text{def}}{=} f \circ B^{\mathsf{T}}$ yields the revised quantities

$$\forall_{x \in \mathbb{R}^n} :\ \nabla \tilde{f}(x) = B \cdot \nabla f(B^{\mathsf{T}} x) \quad \text{and} \quad \mathrm{H}_{\tilde{f}}\big|_x = B\,\mathrm{H}_f\big|_{(B^{\mathsf{T}} x)}B^{\mathsf{T}}.$$

___
[5]This is easy to do at $x_0$ since $(x_0, g(x_0)) = (x_0, y_0)$.

Table 2: Accuracies of CIFAR models at different perturbation lengths

| $\epsilon$ | Dimension | | | | | | | | | |
|---|---|---|---|---|---|---|---|---|---|---|
| | 1 | 2 | 3 | 4 | 5 | 6 | 7 | 8 | 9 | 10 |
| 0 | 95.25% / 90.83% | 95.25% / 90.83% | 95.25% / 90.83% | 95.25% / 90.83% | 95.25% / 90.83% | 95.25% / 90.83% | 95.25% / 90.83% | 95.25% / 90.83% | 95.25% / 90.83% | 95.25% / 90.83% |
| 0.1 | 51.43% / 86.29% | 62.86% / 90.83% | 70.48% / 90.83% | 76.2% / 90.83% | 79.06% / 90.83% | 80.01% / 90.83% | 81.92% / 90.83% | 81.92% / 90.83% | 82.87% / 90.83% | 83.82% / 90.83% |
| 0.25 | 1.91% / 79.93% | 4.76% / 89.01% | 5.72% / 89.92% | 6.67% / 90.83% | 7.62% / 90.83% | 8.57% / 90.83% | 9.52% / 90.83% | 9.52% / 90.83% | 9.52% / 90.83% | 10.48% / 90.83% |
| 0.5 | 0.0% / 65.4% | 0.0% / 84.47% | 0.0% / 86.29% | 0.0% / 87.2% | 0.0% / 87.2% | 0.0% / 87.2% | 0.0% / 87.2% | 0.0% / 87.2% | 0.0% / 87.2% | 0.0% / 87.2% |
| 1 | 0.0% / 36.33% | 0.0% / 54.5% | 0.0% / 59.04% | 0.0% / 61.76% | 0.0% / 65.4% | 0.0% / 69.03% | 0.0% / 69.94% | 0.0% / 70.85% | 0.0% / 72.66% | 0.0% / 73.57% |
| 2 | 0.0% / 2.72% | 0.0% / 5.45% | 0.0% / 10.9% | 0.0% / 12.72% | 0.0% / 13.62% | 0.0% / 14.53% | 0.0% / 15.44% | 0.0% / 15.44% | 0.0% / 16.35% | 0.0% / 17.26% |

## B  DATASET AND MODELS

For the adversarial subspace analysis we used two identical ResNet-50 architectures (He et al., 2016) and two identical WideResNet-70 architectures (Gowal et al., 2020). For each architecture, one model was trained on the unmodified CIFAR-10 dataset (Krizhevsky & Hinton, 2009). For the second model we additionally incorporated adversarial examples in the training process, widely known as adversarial training. A training perturbation size $\epsilon$ of 0.5 was used. To perform the curvature analysis, we used twice-differentiable ELU activation functions (Clevert et al., 2015) for both models, which had little impact on the clean or adversarial accuracies.

In the analysis we include 100 CIFAR-10 test images, class-balanced, all unseen during training, and correctly classified by both models. The optimization algorithm stopped after finding 50 orthogonal adversarial vectors for all 100 test images. For the curvature analysis we used a subset of 50 class-balanced images and 10 adversarial directions per image.

Additionally, we only considered one seed per CIFAR-10 model because we found that the analysis on the MNIST trained models had little variation across seeds (see Appendix E). The accuracy of both models at different perturbation lengths $\epsilon$ and dimensions is shown in Table 2. Here, the first line of each row corresponds to the accuracy of the naturally trained model, whereas the second line is referring to the adversarially trained model.

We repeated many of the experiments on the MNIST dataset (LeCun et al., 1998) and found largely consistent results (see Appendix D).

## C  ADDITIONAL CIFAR-10 RESULTS

Here we show additional analysis to complement our findings on the CIFAR-10 dataset.

Figure 6 shows five adversarially perturbed images and their corresponding original image found for the two ResNet-50 models. We can see that that the perturbation length is increasing with dimension number. Interestingly, while the adversarial categories may not be semantically adjacent to the clean classes, the adversarially trained model seems to require more semantically meaningful perturbations, e.g. the images of boats in the top row, misclassified as cars, are perturbed to have elements looking like wheels. We can also see that there is no clear, consistent $\ell_2$-length of an adversarial attack that makes an image ambiguous for a human observer. For example, the robust model example in the first (ship) row of the adversarial 11 column pair has a perturbation length of 1.12 and could be mistaken as a car on a first glance. In contrast, the robust model example in the fourth (horse) row of the adversarial 1 column pair is still clearly identifiable as a horse with a larger perturbation length of 1.2. This illustrates that the direction of the perturbation is as important as the

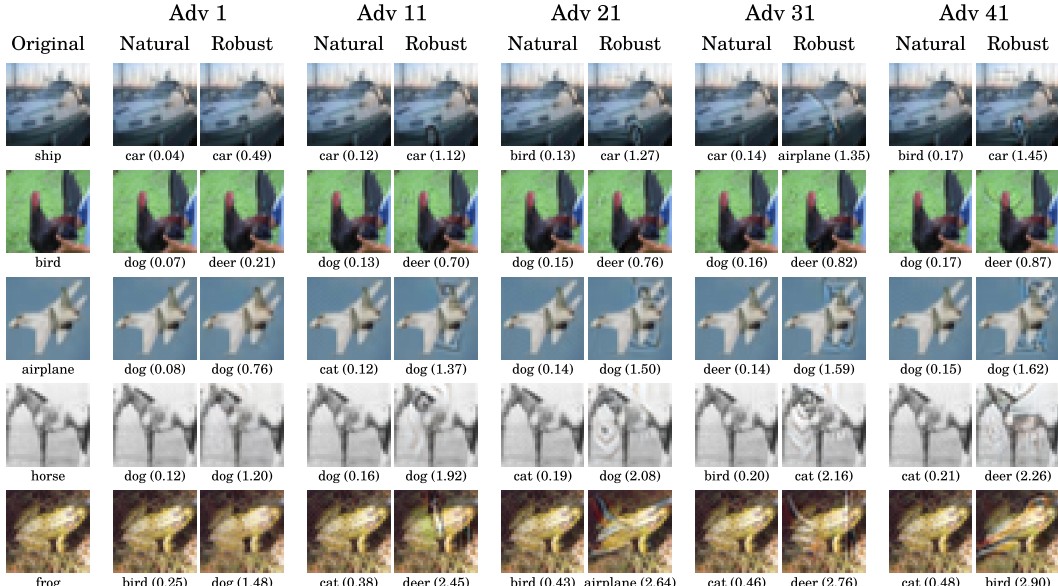

Figure 6: **CIFAR-10 example adversarials**. The panels show adversarial examples for a given input image. Column pairs are sorted by adversarial dimension. Within a pair, each column includes examples for the natural and robust models, respectively. The labels below the images indicate the class assigned by the model. The numbers in brackets are the $\ell_2$-norms of the respective images.

perturbation magnitude itself, as was previously reported by (Goodfellow et al., 2014; Ilyas et al., 2019), although assessing the "perceptual distance" is an open area of research.

As was described earlier, we uniformly randomly sampled directions within adversarial subspaces to measure the distance to the decision boundary. Figure 7 shows how many of the samples pointed to decision boundaries that were outside of the allowed input pixel bounds.

Figure 8 shows that perturbation lengths are increasing sublinearly and monotonically with increasing number of adversarial dimensions. For the naturally trained model there is also a much larger increase of adversarial dimension with increasing distance to distance to decision boundary. This figure is different from e.g. Figure 2 in that it displays the lengths of the adversarial subspace axes, not the lengths of the random perturbations within these subspaces. Note that in the right panel of this figure both models plateau at 50 dimensions because we stopped the optimizer at this point.

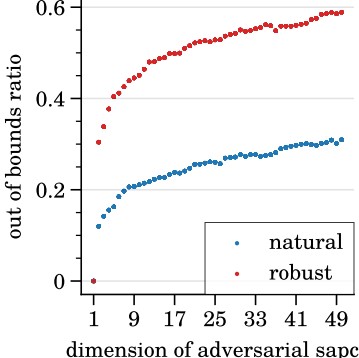

Figure 7: **Ratio of out of bounds samples**. We uniformly sample the distance to the decision boundary with increasing dimensionality per input image and adversarial subspace dimension. We report the fraction of directions pointing to out-of-bounds boundaries, averaged across 100 images.

We also provide additional, and more detailed information on the distributions of distances to the decision boundary of randomly sampled vectors in adversarial subspaces in Figure 9. Each subplot shows a subspace of increasing dimensionality, ascending left-to-right, then top-to-bottom. We observe that the variance of the distances to the decision boundary of randomly sampled vectors is smoothly increasing with dimensionality.

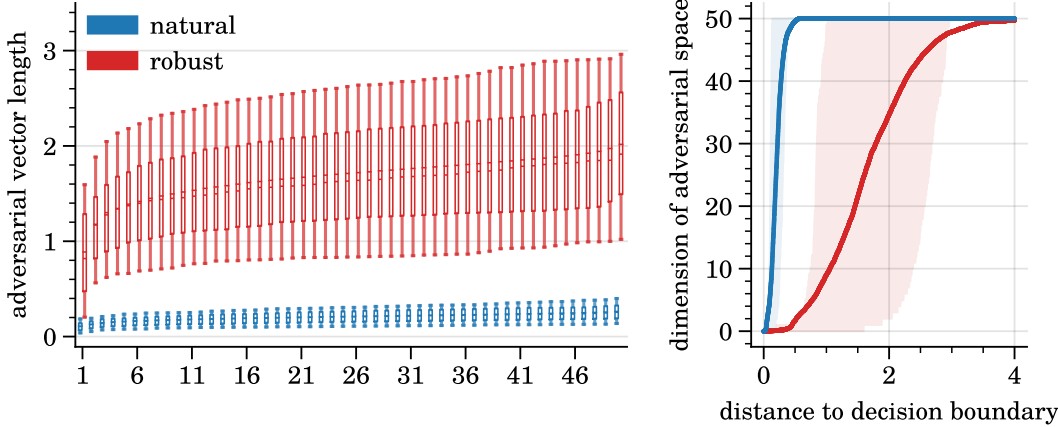

Figure 8: **Perturbation lengths and dimensionality**. (left) The subfigure shows the distribution of the n-th found adversarial perturbations across images. (right) We tested at distance increments of 0.001 how many adversarial dimensions were found at that distance per image. The lines represent the means over all test images and the color shaded areas show the 10th and 90th percentiles.

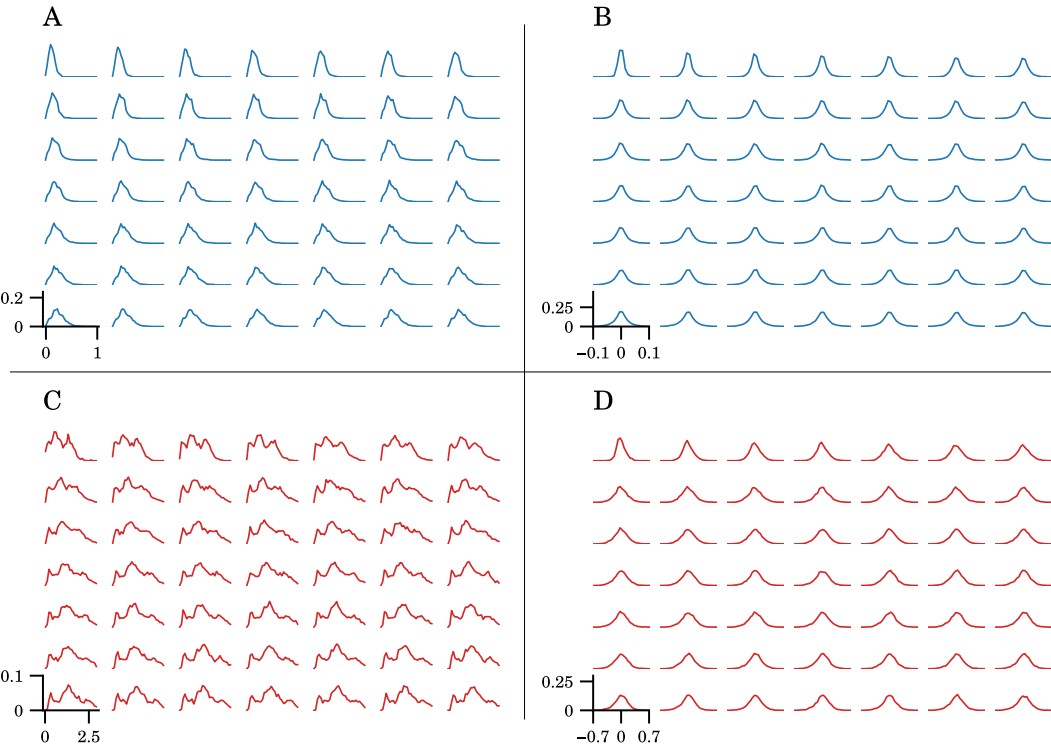

Figure 9: **Distributions of distances to decision boundary**. (A,B) Distributions for natural model. (C,D) Distributions for adversarially trained model. (A-C) We show the complete set histograms corresponding to Figure 2 dimensions. (C,D) Additionally, we provide the mean-normalized histograms to corresponding to the histograms in A and C to emphasize the smooth increase in variance of distances with increasing dimensionality.

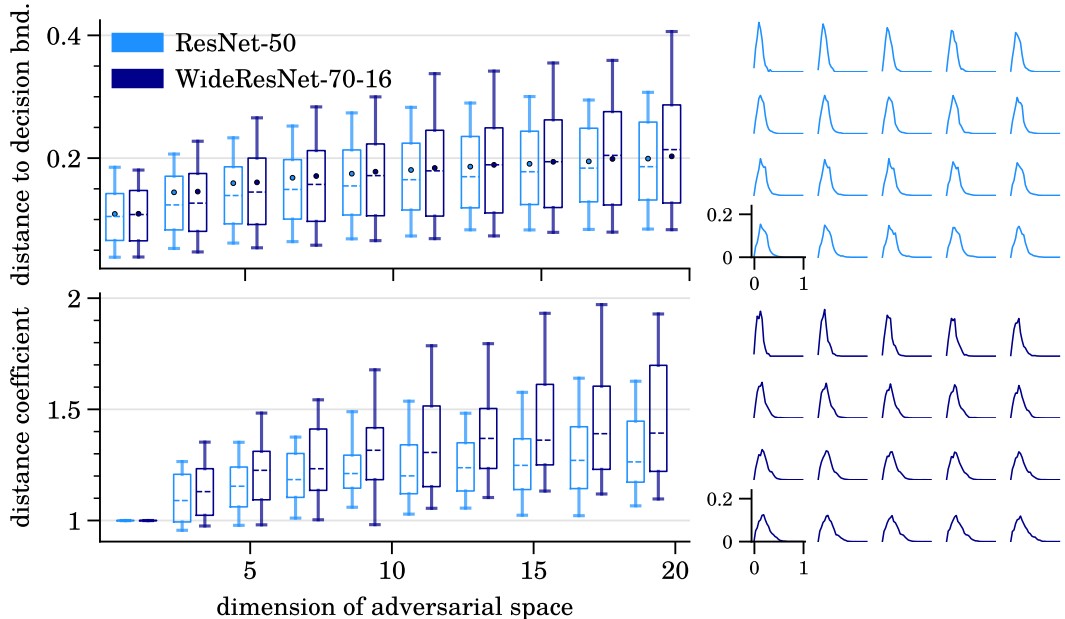

Figure 10: **Distances to the decision boundary**. Compare with Figure 2.

ADDITIONAL MODEL COMPARISON

To show that our findings hold for different models, we present results for two different architectures: a ResNet-50 (He et al., 2016) and a WideResNet-70-16, of which the latter is the state-of-the-art adversarially trained network for CIFAR-10 images (Gowal et al., 2020). Both architectures were trained with and without adversarial perturbations, resulting in 4 models overall. Figures 10 through 14 repeat analysis to compare natural and adversarial trained models, but instead compare models of different architectures and constant training procedure.

Figures 10, 11, and 14b show a comparison of the two naturally trained models. These figures indicate that there is little difference between the two model architectures regarding distance to the decision boundary and perturbation length over dimensions. Figures 12, 13, and 14a show a comparison of the two adversarially trained models. Here, the figures show that perturbation lengths as well as distances to the decision boundary are larger for the WideResNet-70-16 compared to the ResNet-50. This is expected, as the WideResNet has higher adversarial accuracy and thus larger perturbations are needed to fool it. At the level of these aggregate statistics, the hyperparameter differences in adversarial training (Figure 12) have more of an effect than the model architecture differences themselves (Figure 10), although all differences are much less pronounced than the differences between adversarially and naturally trained networks (Figure 2).

The box plots depicting the distances to the decision boundary in Figures 10 and 12 show that, for both architectures, the adversarial subspace is large, continuous, and increases sublinearly in volume as the dimensionality is increased. The observation that this finding is agnostic to model architecture is further supported by the similarity in distributions of samples measuring the distance to decision boundary that can be seen on the right hand side of the same figures. Thus, the observed trends and conclusions drawn from the analysis provided in this study hold across architectures.

# D  ADDITIONAL MNIST RESULTS

We conducted the same experiments described in Section 3 on networks trained with the MNIST dataset. As a test set we used 500 images (50 per class), unseen in training and correctly labeled by all investigated models.

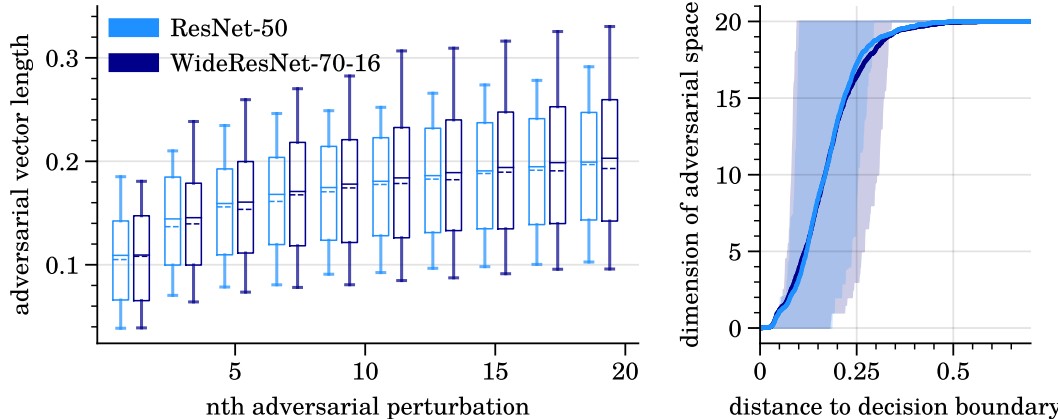

Figure 11: **Perturbation lengths and dimensionality**. Compare with Figure 8.

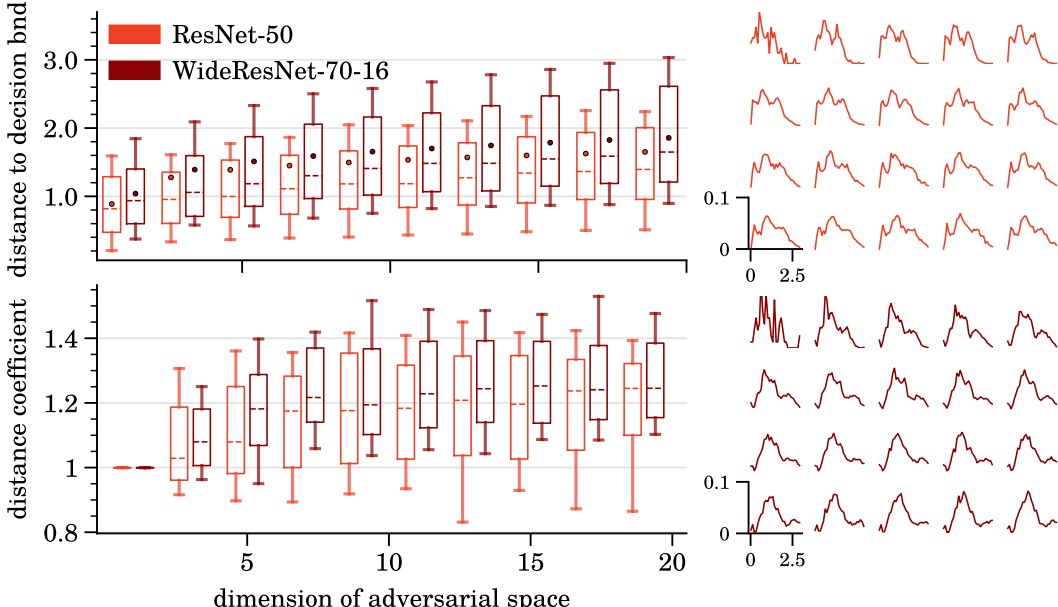

Figure 12: **Distances to the decision boundary**. Compare with Figure 2.

In Figure 18 we can see that perturbation lengths are much larger for MNIST than for CIFAR-10, although the relative increase in distances for the robust network is still present. The optimization was in general more difficult with MNIST, likely due to the fact that the images all live very close to the allowable input bounds. Specifically, the algorithm found less than 30 adversarial dimensions for more than 95% of the test images (Figure 22), to be compared with finding >50 dimensions for 100% of the CIFAR-10 test images.

That it is harder to fool the MNIST model can also be seen in Table 3: the accuracies of both models are much higher for the same perturbation lengths than the accuracies of the CIFAR-10 models. For the curvature analysis, on the other hand, we found that the overall values were larger on CIFAR-10. Additionally, the adversarially trained model had more decision boundary curvature within adversarial subspaces on MNIST than on CIFAR-10.

We noticed that most of the MNIST adversarials lie in a thin regions between the decision boundary and the valid pixel box boundary (compare Figure 17 against Figure 3). Figure 15 shows that this leads to large rates of out of bounds samples when measuring the distance to the decision boundary. For example, 90% of the time the decision boundary is outside the valid pixel range in the 6 dimensional adversarial subspace of the naturally trained model.

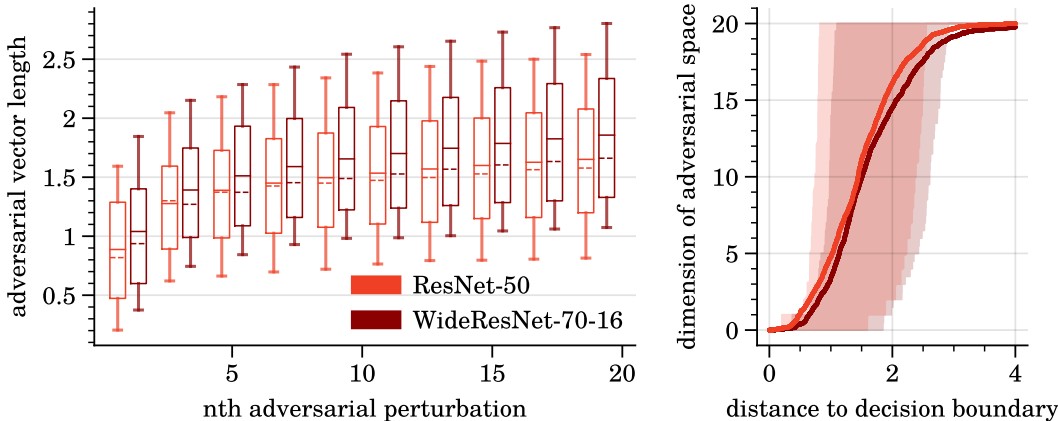

Figure 13: **Perturbation lengths and dimensionality**. Compare with Figure 8.

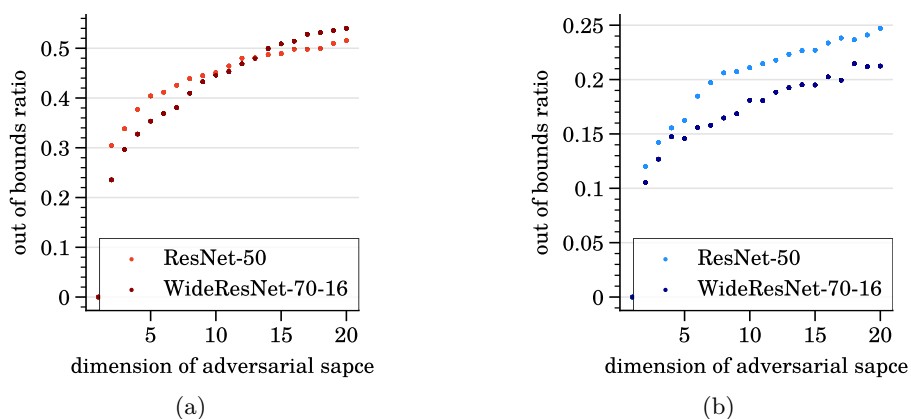

(a)                                              (b)

Figure 14: **Ratio of out of bounds samples**. Compare with Figure 7.

Table 3: Accuracies of MNIST models (seed 0) at different perturbation lengths

| $\epsilon$ | Dimension | | | | | | | | | |
|---|---|---|---|---|---|---|---|---|---|---|
| | 1 | 2 | 3 | 4 | 5 | 6 | 7 | 8 | 9 | 10 |
| 0 | 98.59% / 98.56% | 98.59% / 98.56% | 98.59% / 98.56% | 98.59% / 98.56% | 98.59% / 98.56% | 98.59% / 98.56% | 98.59% / 98.56% | 98.59% / 98.56% | 98.59% / 98.56% | 98.59% / 98.56% |
| 0.5 | 90.51% / 97.77% | 97.41% / 98.56% | 98.0% / 98.56% | 98.39% / 98.56% | 98.39% / 98.56% | 98.39% / 98.56% | 98.37% / 98.56% | 98.35% / 98.56% | 98.29% / 98.56% | 98.23% / 98.56% |
| 1 | 60.53% / 94.03% | 90.31% / 98.36% | 95.83% / 98.56% | 97.01% / 98.56% | 97.4% / 98.56% | 97.58% / 98.56% | 97.94% / 98.56% | 97.86% / 98.56% | 97.69% / 98.56% | 97.51% / 98.56% |
| 2 | 1.77% / 68.99% | 18.34% / 89.3% | 47.91% / 95.01% | 71.58% / 96.59% | 83.54% / 97.57% | 88.51% / 97.69% | 91.63% / 97.73% | 93.0% / 97.72% | 93.8% / 98.56% | 93.55% / 98.56% |
| 4 | 0.0% / 0.59% | 0.0% / 9.46% | 0.2% / 27.99% | 4.14% / 46.52% | 15.05% / 65.77% | 26.41% / 76.32% | 32.43% / 82.59% | 38.61% / 83.94% | 37.46% / 85.42% | 32.74% / 80.16% |
| 6 | 0.0% / 0.0% | 0.0% / 0.0% | 0.0% / 0.2% | 0.39% / 7.69% | 1.98% / 24.69% | 7.86% / 37.51% | 10.45% / 47.63% | 18.94% / 48.86% | 16.48% / 54.54% | 16.55% / 40.74% |

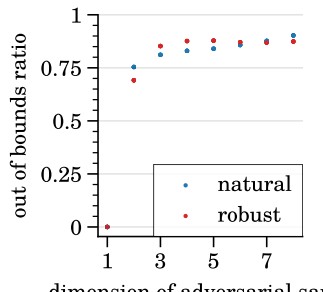

Figure 15: **Ratio of out of bounds samples**. Compare to Figure 7.

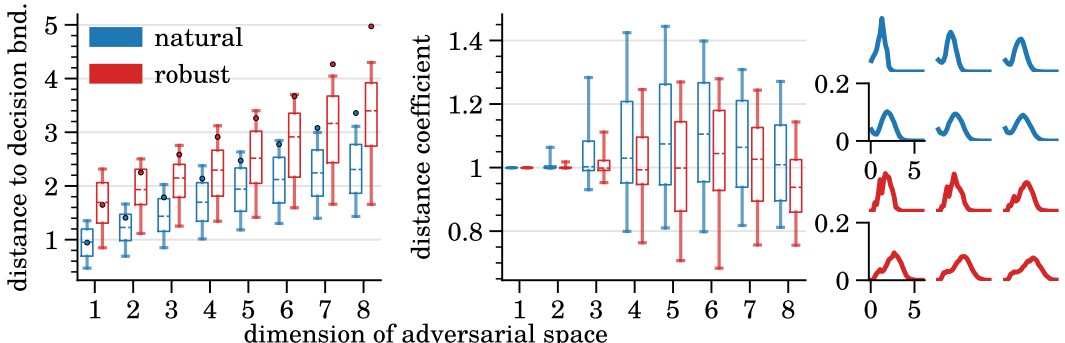

Figure 16: **MNIST distances to the Decision Boundary.** We repeated the distance analysis from Figure 2 on the MNIST dataset Only test images with at least 8 adversarial dimensions were included in this Figure

## E  Seed consistency

We trained the natural and robust MNIST models with five different seeds and extracted adversarial subspaces for each model to examine the influence of training seeds on our results. Figure 22 shows that there is slight variation in number of adversarial subspace directions found by the optimizer. However, the overall trend of bar heights is similar across seeds and there are no significant outliers.

To further validate our results, we repeated the experiment for Figure 16 (measuring the length of the adversarial vector as one increases dimensionality on the MNIST trained networks) across five different seeds per training type. The results are shown in Figure 23. The variation across seeds for a

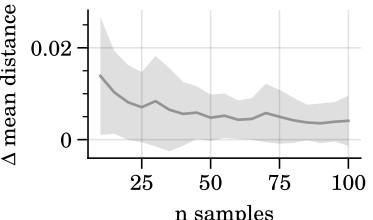

Figure 21: **Sampling size in adversarial subspaces**. The line shows the mean running difference in distance to the decision boundary means across 100 samples as one increases the sample size from 5 to 100. The shaded area depicts the standard deviation from the mean line.

single image is smaller than the variation across images for a single seed, which can be seen by comparing Figure 23 to Figure 16. In general, we also found that the results and trends were consistent between MNIST and CIFAR-10 networks (with differences noted in Appendix D). Thus, we considered one seed per CIFAR-10 model as sufficiently representative.

We also investigated the influence of random number generation relevant for sampling within *adversarial subspaces*. As an empirical test, we varied the number of samples and measured the change in mean distance to the decision boundary as one increases the number of samples. This was repeated for 100 different 50-dimensional subspaces (i.e. subspaces for 100 different test images) with the naturally trained CIFAR network. Figure 21 shows that the absolute change decreases steadily to an average l2 distance of 0.004. Thus, the mean variation from 100 samples is much smaller than the general distance scales we are measuring throughout

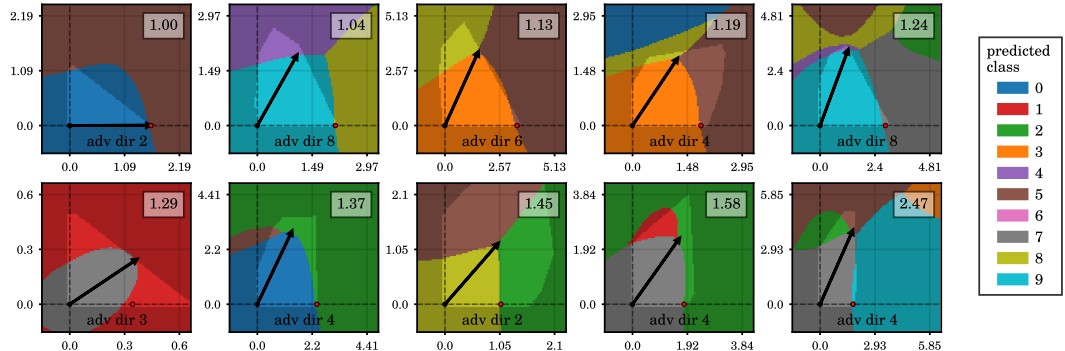

Figure 17: **MNIST boundary visualizations.** This figure is the MNIST version of Figure 3. Note that for MNIST, the decision boundary lies outside the valid pixel range more often than for CIFAR.

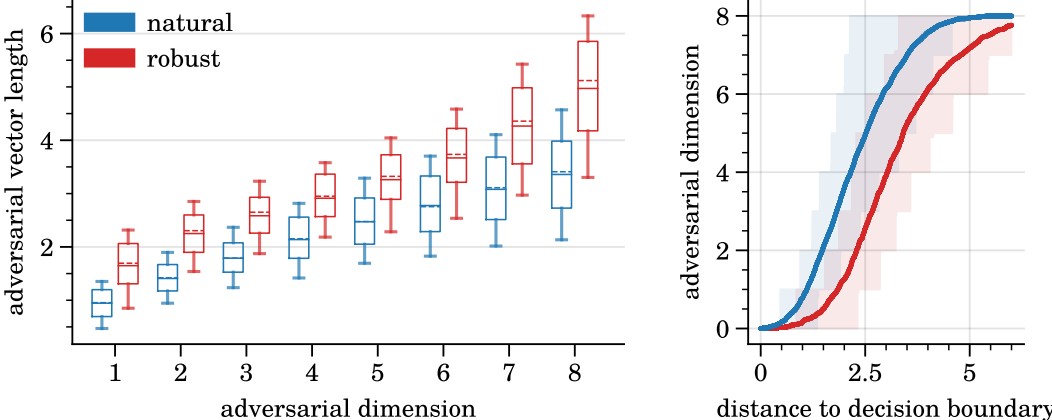

Figure 18: **MNIST perturbation lengths and dimensions.** Compare with Figure 8.

the paper. We also measured the variation in distances across random seeds for the sampling process. To do this, we used 10 different seeds to generate 100 samples in a 50-dimensional subspace corresponding to one test image. For each seed, we compute a mean distance, averaged across 100 samples, giving us 10 values. The mean distance across seeds was 0.0933, and the standard deviation was 0.0007. The standard deviation is less than 1% of the mean and thus sufficiently small to consider an estimate with 100 samples as representative.

## F  COMPARISONS TO RELATED WORK: ORTHOGONALITY AND CLIPPING

One obstacle in finding proper adversarial subspaces spanned by *orthogonal* vectors is assuring that perturbations do not cause pixel values to be out of the valid input boundaries $[0, 1]$. We noticed that many studies solve this problem by element wise pixel clipping. Here, we show that pixel clipping is problematic when talking about the dimensionality of subspaces. Element wise pixel clipping is a non-linear operation performed on a vector and can therefore change its direction in the input space and its relative angle to other, fixed vectors.

To show that clipping has a non-negligible effect, we investigated relative angles of basis vectors that span (sub-) spaces for 100 MNIST test images relevant for two studies.

First, in their work, Tramèr et al. (2017) propose the GAAS method. GAAS estimates the dimensionality of adversarial subspaces by defining an orthogonal set of vectors aligned with an attack vector to an input image. However, before passing the orthogonal basis vectors to the model, a clipping operation is performed. Figure 24 (left) shows that this clipping operation causes the input vectors to be non-orthogonal – orthogonality would constitute an angle of 90° or $\frac{\pi}{2}$. In fact, for the 100 sample images we tested, there is not a single vector pair that is completely orthogonal. The author's conclusions regarding the dimensionality

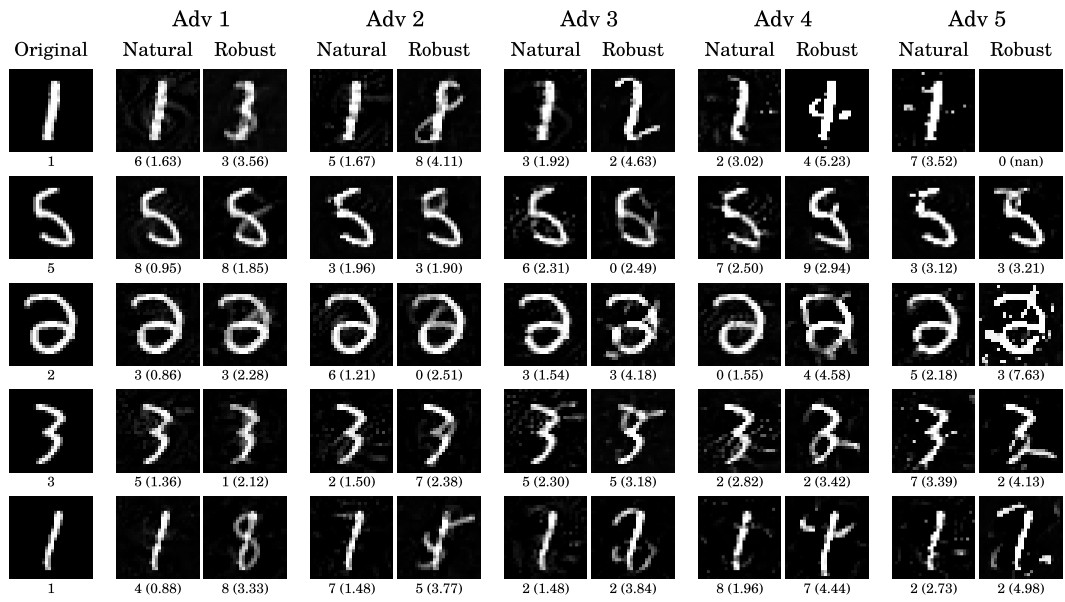

Figure 19: **MNIST example adversarials.** Compare to Figure 6.

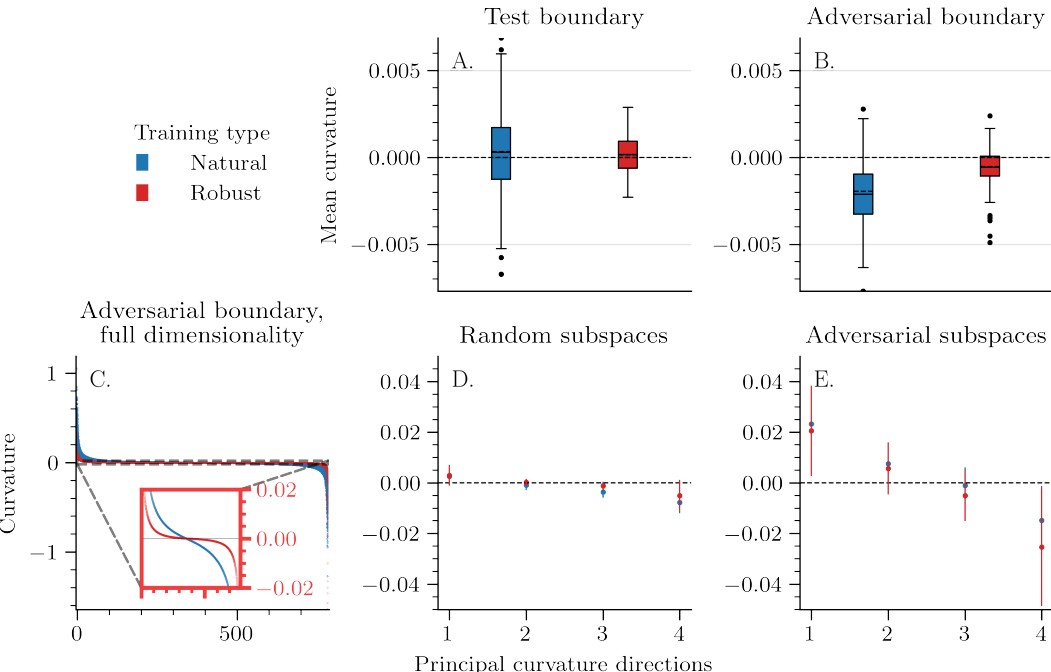

Figure 20: **MNIST decision boundary curvature.** Compare with Figure 4

of adversarial subspaces and transferable subspaces are therefore grounded on ill-defined subspaces, spanned by non-orthogonal vectors.

In another study, He et al. (2018) estimate the distance to the decision boundary for the complete input space of MNIST. For that, a basis of 784 vectors, spanning the whole input space, is randomly generated. Again, the inputs are clipped before they are passed to the model leading to the same problem as described earlier. And as before, there are no orthogonal vector pairs, and angels are accumulated at much smaller values than $\frac{\pi}{2}$ as can be seen in Figure 24 (right). We therefore note that it is not guaranteed that the input space is evenly covered and that edges of the 784-dimensional hypercube (in the case of MNIST) are arguably overrepresented.

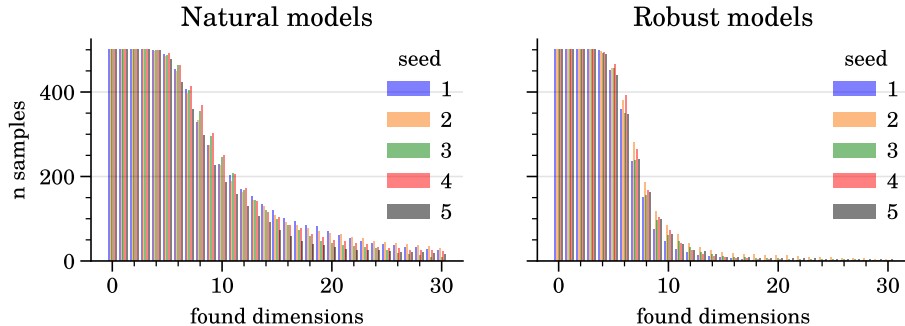

Figure 22: **Optimization consistency on MNIST**. The bars show number of samples for which at least n dimensions were found during optimization. Different colors indicate distinct seeds for model training.

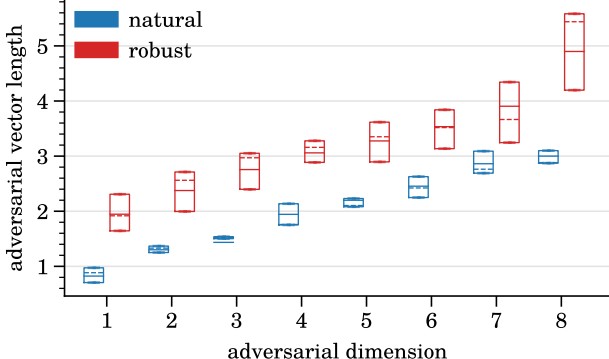

Figure 23: **Distance consistency on MNIST**. Box plots show the length of the adversarial vector as one increases dimensionality. The data is computed for a single image, where the variation comes from five different seeds used to initialize the networks before training. Note that the variation is much smaller than the variation across images for a single seed, as can be seen in Figure 16.

## G    COMPARISONS TO RELATED WORK: DECISION BOUNDARY CURVATURE

Table 4: Decision boundary curvature formulas present in the literature for a decision function $F_{ij} : \mathbb{R}^n \to \mathbb{R}$. The *generating matrix* column refers to the matrix used to yield eigenvalues for the principle curvatures, and is roughly comparable to the shape operator.

| Reference | Derivation | Generating matrix | $P$ | Dimension |
|---|---|---|---|---|
| Poole et al., 2016 | None | $\frac{P\,\mathrm{H}_{F_{ij}}\,P}{\lvert\nabla F_{ij}\rvert_2}$ | $\mathrm{Id} - \frac{\nabla F_{ij} \nabla F_{ij}^{\mathsf{T}}}{\lvert\nabla F_{ij}\rvert^2}$ | $n \times n$ |
| Moosavi-Dezfooli et al., 2018 | None | $P\,\mathrm{H}_{F_{ij}}\,P$ | Not given | $n \times n$ |
| Moosavi-Dezfooli et al., 2019 | None | Crossenetropy Hessian | - | $n \times n$ |
| Moosavi-Dezfooli, 2019, p. 90 | None | $\frac{P\,\mathrm{H}_{F_{ij}}\,P}{\lvert\nabla F_{ij}\rvert_2}$ | $\mathrm{Id} - \nabla F_{ij}\nabla F_{ij}^{\mathsf{T}}$ | $n \times n$ |
| This work | Appendix A | (11) using (15) | - | $n-1 \times n-1$ |

This section provides a detailed comparison of our analysis against previously published works on the principal curvature analysis applied to decision boundaries. We include table 4 for a quick summary.

The use of differential geometry to understand the non-linear response properties of model neurons has a long history in computational neuroscience, with early contributions made by Zetzsche & Barth (1990). More recently, several works have specifically used Reimannian geometry to measure the principal curvature of response manifolds (Fawzi et al., 2016; Poole et al., 2016; Moosavi-Dezfooli et al., 2018; 2019; Golden et al., 2019). As we discuss in

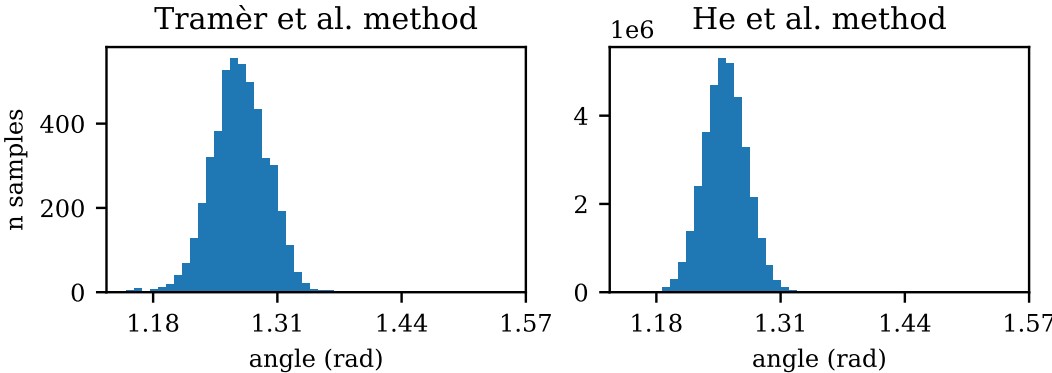

Figure 24: **Clipping causes non-orthogonality**. The histograms show angles between vectors that define "subspaces" in the GAAS method proposed by Tramèr et al. (2017) (left) and distance to decision boundary estimations by He et al. (2018).

Appendix A, certain smoothness assumptions are required for such an analysis. Our present study, as well Poole et al. (2016) and Golden et al. (2019), avoid this problem by using twice-differentiable activation functions. In the case of Golden et al. (2019), this was done without retraining the network, while Poole et al. (2016) used nets with random weights. We instead trained the twice-differentiable networks from scratch. Moosavi-Dezfooli et al. used a finite-difference approximation to avoid the problem (described in Moosavi-Dezfooli et al., 2019, , eq 1 and surrounding discussion). Another high-level difference, discussed in detail in Appendix A, is the choice of measuring the response graph curvature (which is $n$ dimensional for $n$ dimensional inputs) or the level-set curvature (which is $n-1$ dimensional for $n$ dimensional inputs) of a given function. This difference was discussed in (Golden et al., 2019, , at the end of Section 4 and in Figure 4), although their technical derivations are limited only to the graph formulation. The other studies did not explicitly discuss this difference. Importantly, though, the decision boundary is the level-set of the logit difference function, as defined in Equation 2.

Several works from Moosavi-Dezfooli and colleagues explore decision boundary curvature and its relationship to adversarial robustness (Fawzi et al., 2016; Moosavi-Dezfooli et al., 2017b; 2018; 2019; Moosavi-Dezfooli, 2019). Within these works they provide an inspired perspective on explaining adversarial robustness in terms of the curvature of decision boundaries. Almost all of the work is with respect to *universal* adversarial perturbations, which share a common subspace (Moosavi-Dezfooli et al., 2017a), although Fawzi et al. (2016) alternatively look at random subspaces. Their universal adversarial perturbation subspaces differ from our adversarial subspaces in that theirs are defined by orthogonal directions that transfer well across inputs instead of those that point to the nearest decision boundary for a given input. Despite these differences, the stated quantity of interest, that is, the principle curvatures at a point on the decision boundary is the same in their work as well as ours.

However, each of these papers have a slightly different analytic formula for estimating the curvature of the boundary within a given subspace (Fawzi et al., 2016; Moosavi-Dezfooli et al., 2017b), or in the high-dimensional input space (Moosavi-Dezfooli et al., 2018; 2019). We found the most complete description to be given in Moosavi-Dezfooli (2019), which summarizes a majority of the earlier work and applies the decision boundary curvature analysis to both subspaces and the high-dimensional input space. Therein, Moosavi-Dezfooli (2019, Section. 7.4) state the following formula, without derivation, for the shape operator (in the notation of Section 2.2):

$$s = \frac{1}{|\nabla F_{ij}(x)|_2} P \, \mathrm{H}_{F_{ij}} \, P,$$

where $\mathrm{H}_{F_{ij}}$ is the Hessian of the boundary function, $F_{ij}(x) \stackrel{\text{def}}{=} f_i(x) - f_j(x)$, and $P \stackrel{\text{def}}{=} \mathrm{Id} - \nabla F_{ij}(x) \nabla F_{ij}(x)^{\mathsf{T}}$ is described as a projection operator. This formulation is similar to that used by (Poole et al., 2016), with the only difference being that they use normalized gradients in the projection operator.

Table 5: Calculations from alternative attempts find incorrect principal curvature for a level-set on a sphere of varying dimensionality and radius. The quantity reported is the mean of the absolute difference between each computed principal curvature and the inverse radius.

| dimensionality-radius | Moosavi et al. | Poole et al. | Ours |
|---|---|---|---|
| 02 - 00.1 | 7.6e-01 | 5.0e+00 | 0.0e+00 |
| 02 - 01.0 | 2.4e+01 | 5.0e-01 | 0.0e+00 |
| 02 - 10.0 | 3.1e+04 | 5.0e-02 | 1.3e-17 |
| 05 - 00.1 | 7.2e-01 | 2.0e+00 | 2.2e-15 |
| 05 - 01.0 | 7.2e+01 | 2.0e-01 | 2.2e-16 |
| 05 - 10.0 | 7.9e+04 | 2.0e-02 | 3.8e-17 |
| 10 - 00.1 | 6.4e-01 | 1.0e+00 | 2.3e-15 |
| 10 - 01.0 | 1.5e+02 | 1.0e-01 | 2.2e-16 |
| 10 - 10.0 | 1.5e+05 | 1.0e-02 | 4.4e-17 |

Table 6: Calculations from alternative attempts find incorrect Gaussian curvature (i.e. the product of the principal curvatures) for a Hyperboloid graph.

| Analysis type | Absolute error |
|---|---|
| Moosavi et al. | 6.8e+00 |
| Poole et al. | 1.4e-03 |
| Golden et al. | 4.1e-10 |
| Ours | 4.1e-10 |

Our formulation differs from the above in several ways, the most important of which is that they claim to transform the Hessian by projecting it onto the tangent space (as written above), while we apply the implicit function theorem to parameterize the decision boundary as the graph of a function (see Appendix Section A.2.2, Equation 13). For computing the curvature of the decision boundary, exclusively projecting the Hessian is not going to result in the correct calculations, since for the decision boundary (as defined in Section 2.2, Equation 2) you need a different normal than for the graph of the response function. More specifically, the normal used for computing the shape operator of the graph surface is different from that for computing the shape operator of the decision boundary. The latter has to be exclusively in the pixel space, while the former always has a component in the direction of the function response. Projecting the Hessian can restrict the computed curvatures in a certain subspace of directions, but it cannot change the computed curvatures in said subspace. Another important difference is their projection operator is computed directly from the gradient, and thus has a dimensionality equal to the input dimensionality. To calculate the curvature of a decision boundary, one needs a level-set formulation, which should have one fewer dimensions than the input space.

The consequence of this difference can be made clear when looking at the curvature of an $n$-dimensional sphere, as defined by the quadratic function:

$$F(x) = x^T H x,$$

where $x$ is the input point vector and $H$ is the sphere's Hessian (i.e. an identity matrix in $\mathbb{R}^{n \times n}$). Given this function, the principal curvatures of the level-set at some radius, $r$, are all known to be (up to a sign change) $\frac{1}{r}$. However, we show in table 5 that the formulations provided by Moosavi-Dezfooli (2019) and Poole et al. (2016) produced curvatures that are often several orders of magnitude off from the correct values the case of high-dimensional inputs. Due to the dimensionality issue above, we computed the average error across each principal curvature, regardless of the number of curvatures reported, as they should all equal $\frac{1}{r}$.

We additionally tested a graph formulation (defined in Section A.2.1), using a one-sheeted hyperboloid. This formulation lives in the full space, so there is a principal curvature per input dimension. However, we show similar errors in in Table 6. Note, that the formulation from (Golden et al., 2019) matched our own, although they did not provide an equation

for mapping this to the level-set curvature. Our own implementations were correct with high precision on these and other experiments with known functions, all of which can be replicated from our provided code. Finally, while the subspace projection methods proposed by (Fawzi et al., 2016; Moosavi-Dezfooli et al., 2017b) potentially provide metrics related to curvature in the subspaces they defined, we chose to instead derive our pullback method described in Equation 3 from primary sources, as explained in Appendix A.

