# OpenReview forum: "The Geometry of Adversarial Subspaces"
_ICLR.cc/2022/Conference — ICLR 2022 Submitted_

### Official Review · Reviewer_xfHm · 2021-10-29

**Correctness:** 3
**Technical Novelty And Significance:** 3
**Empirical Novelty And Significance:** 3
**Recommendation:** 6
**Confidence:** 4

**Main Review:**

The strengths of the paper are:
- fully taking into account the boundedness of the input domain when finding adversarial subspaces
- principled and (almost) fully rigorous way of defining and measuring the curvature of the decision boundary
- empirical investigation of the geometry of the decision boundary, leading to interesting (although not shocking) results

I have two doubts about the paper, which might require more discussion in the paper (or I didn't understand something):
1. Riemannian geometry machinery requires a smooth (i.e., infinitely differentiable) manifold. We know that the activation functions used in typical neural network models are not smooth. Hence, the decision boundaries will not be smooth either (but will be locally smooth). How does that affect the results? Is the local curvature representable of the large-scale geometry of the differentiable manifold if the manifold is not smooth? A zig-zagging line is locally linear almost everywhere, but can curve in space as much as it wants. If we sample its local curvature randomly, we will "see it" as a straight line.
2. There is a lot of space in 50 dimensions. Is sampling e.g. 100 random vectors in a 50-dimensional adversarial subspace going to give us a representative sample? Have the authors tried to repeat the analyses with different random seeds and see if the empirical results are stable?

The most important doubt is 1. I think it invalidates the results of the curvature analysis, viz the example of the zig-zag line. Since this analysis is a major selling point of the paper, I cannot recommend accepting it at this stage.

Some other comments:
1. I think the paper "Visualizations of Decision Regions in the Presence of Adversarial Examples" (Swirszcz, O'Donoghue and Kohli) should also be referenced as it is related work.
2. The first sentence in the caption for Figure 1 is hard to understand.
3. In the last sentence of the caption for Figure 2, what is "even subsampling"?
4. What is meant by "well approximated" in the discussion of the modification to the optimisation problem described by Eqn. (1)? Can the Authors provide some proof that this approximation works?
5. I think the folk lore is that in order to prevent the perturbations from being visible to human viewer, it's better to cap their L_inf norm, not L_2. Something to consider.

==== EDIT AFTER AUTHOR REPLY ====

The Authors have completely clarified Q2 (representative samples) to my satisfaction. Re Q1 (smoothness), they explained that they use a differentiable (though not smooth) ELU activation function, which alleviates my prior concerns, but raises the question of the applicability of their results to the more common case of ANNs using non-differentiable activation functions like RELU. However, this may be seen as an open question to be explored in future work. (It would also be nice if the Authors reviewed prior work for curvature analysis and commented on whether it also avoided the problem of using non-differentiable network architectures).

**Summary Of The Paper:**

The paper investigates the geometry of the decision boundary surrounding test examples which are subjected to successful adversarial attacks. It uses Riemannian geometry to calculate the curvature of the boundary in a principled way. It also finds "adversarial subspaces" in the input space, which are spanned by orthogonal vectors pointing from the test example to the closest adversarial examples. Improving on prior work, the Authors ensure that the spanning vectors remain orthogonal after clipping them to ensure that the perturbed examples remain in the domain of the classifier (i.e., the [0, 1]^N hypercube). The empirical results are obtained by randomly sampling the adversarial subspaces. Using this method, the authors investigate the impact of adversarial training on the decision boundary: its distance from test examples and its curvature.

**Summary Of The Review:**

I think that the paper is well-written, continues an interesting line of research and improves on it. The Authors clarified the technical questions. While I'm not sure how applicable the results are to common ANN architectures (because of the smoothness requirements), I recommend acceptance.

---

> ### Author Response · Authors · 2021-11-12
> **Network smoothness & sampling convergence**
>
> Dear reviewer xfHm,
>
> Thank you for taking the time to review our submission and finding the work interesting. We would like to offer an initial reply to address your biggest concerns (correspondingly numbered below). We will follow up with additional experiments and responses to your remaining points in the coming days.
>
> 1 -- As you observed, our theoretical analysis (detailed in Appendix A of the latest PDF) assumes a smooth manifold. With this in mind, we looked at neural networks with smooth activation functions (ELU, which is a common choice for adversarial learning (Gowal et al., [2020](https://arxiv.org/abs/2010.03593))). We explained this in the appendix, but it was not made explicit in the main paper -- in our next update we will make sure this is explicit in the methods section.
>
> 2 -- Your secondary concern was that the number of random boundary samples within the adversarial subspaces might not be enough for our results to be representative. We fully agree with you that properly sampling a high-dimensional space can be difficult and we suspect other readers will share your concern. As an empirical test, we varied the number of samples and measured the change in mean distance to the decision boundary as one increases the number of samples. This was repeated for 100 different 50-dimensional subspaces (i.e. subspaces for 100 different test images) with the naturally trained CIFAR network. The absolute change decreases steadily to an average l2 distance of 0.004. Thus, the mean variation from 100 samples is much smaller than the general distance scales we are measuring throughout the paper. You can find the result via this link ([https://ibb.co/YbVH5mh](https://ibb.co/YbVH5mh); Figure 21), which we will include in an updated manuscript.
>
> You also suggested that we measure the variation in distances across random seeds for the sampling process. To do this, we used 10 different seeds to generate 100 samples in a 50-dimensional subspace corresponding to one test image. For each seed, we compute a mean distance, averaged across 100 samples, giving us 10 values. The mean distance across seeds was 0.0933, and the standard deviation was 0.0007. We will also include this in the updated manuscript
>
> We will add the following paragraph to our methods section to clarify the precautions we took to ensure a representative measure of boundary distance within the subspaces:
> >Once the adversarial vectors are found, we conducted experiments to measure the distance to the decision boundary in the space spanned by the vectors (i.e. within the adversarial subspace). The experiment was repeated on 100 test images that were chosen to have equal label representation (10 images per label) and to be correctly classified by both networks. For each test image we found 50 orthonormal adversarial vectors, which allowed us to define a sequentially growing list of adversarial subspaces with dimensionality increasing from 1 to 50. For a given image and subspace we sampled 100 directions via a linear combination of normally distributed variables, which has been previously proven to provide an even sampling (Muller, [1959](https://dl.acm.org/doi/10.1145/377939.377946)). Finally, for each random direction we performed a binary search to find the decision boundary.
>
> In general we found that the results & trends were consistent as one increased subspace dimensionality and across images (error bars in figs 2, 10, 12, 16), which indicates to us that the number of samples is sufficient. Finally, while we’re on the subject, we would like to point you to Figures 22 and 23 (Fig 23 is new) in our latest updated manuscript, where we include control tests across model seeds as well.
>
> We would appreciate it if you would let us know whether the above clarifies those issues and changes your assessment of the significance of our work.
>
> _Edit: Updated figure and appendix references._

---

> > ### Comment · Reviewer_xfHm · 2021-11-12
> > **Thank you for the clarifications**
> >
> > Dear Authors,
> >
> > Thank you for the clarifications.
> >
> > Re 1: This is the information I was missing. The method makes much more sense now. I think the follow-up question is how representative your results are of the more typical case of ANNs which use non-differentiable activation functions? This seems to be a major limitation of the method if your results are to shed light on the vulnerability of standard image classification models to adversarial attacks. I think answering this question would be very valuable and make the paper more interesting to the reader.
> >
> > Re 2: That's excellent, thank you for the detailed analysis.
> >
> > I will increase the score, but I'm still concerned about the applicability of your results to the typical case of ANNs using RELU activation function.

---

> ### Author Response · Authors · 2021-11-12
> **Additional clarifications on related work & replying to remaining comments**
>
> Dear xfHm,
>
>
> We do agree that extending our results to non-differentiable neural networks is important and highly relevant to the field. While this is, in general, an unsolved problem ( see Moosavi-Dezfooli et al. ([2019](https://openaccess.thecvf.com/content_CVPR_2019/papers/Moosavi-Dezfooli_Robustness_via_Curvature_Regularization_and_Vice_Versa_CVPR_2019_paper.pdf)), Eq 1 and the surrounding discussion for what we believe is the closest attempt), many other works (mentioned below) make similar smoothness assumptions. Regardless, we believe that our existing results are still important for the field, as twice-differentiable activation functions are becoming increasingly popular (Gowal et al., [2020](https://arxiv.org/pdf/2010.03593.pdf)). Additionally, note that all of our analyses excluding the curvature analysis (i.e. all main paper figures except Figure 4) are readily applicable to non-twice-differentiable (e.g. ReLU) neural networks.
>
> Regarding reviewing prior work. Please refer to Appendix G of our updated submission for a detailed discussion on prior work for measuring neuron response curvature. All the works mentioned there make similar smoothness assumptions. Poole et al. (2016) and Golden et al. (2019) avoided the problem by using twice-differentiable activation functions. In the case of Golden et al., this was done without retraining the network, while Poole et al. used nets with random weights. Moosavi-Dezfooli et al. used a finite-difference approximation (see link above) to avoid the problem. We instead trained the ELU networks from scratch, and we motivated all of our measurable quantities with proper derivations (Appendix A).  Perhaps more significantly, our best attempt at reproducing their calculations (including using their code when available) resulted in incorrect level-set curvature measurements on simple manifolds.
>
> We would like to additionally address your remaining comments:
>
> 1 -- Thank you for pointing us to this work. It led us to find a few other relevant alternative methods for visualizing 2D cross-sections of the decision space. We added the references to the introduction. The primary difference is that none of those works use alternative adversarial directions for the orthogonal axis of the cross section. Therefore, our visualizations are more relevant due to the fact that they are designed to show where the boundary is closest (as opposed to the boundary distance in random orthogonal directions, for example).
>
> 2 -- Would this be more clear: “For any given input example, we define a subspace where the boundary is closest to the input.”?
>
> 3 -- We removed this confusing sentence. We simply meant to say that we only plotted data for every other increment in dimensions (e.g. subspaces of size 2, 4, 6, …, 50) instead of every increment. This was to make the plot readable.
>
> 4 -- This approximation is standard in the field (e.g. Szegedy, 2014; Tabacof, 2016; Finlay, 2019). You can see that it is working by looking at tables 2 and 3 in the appendix, which report adversarial accuracies for the natural and robust networks. These accuracies are at par with typical scores reported (e.g. compare the leftmost column of our tables to the results [here](https://github.com/MadryLab/robustness)). That all being said, we agree that it is poor form to assert that it is “well approximated” without any support. We will rephrase to say “is commonly approximated”.
>
> 5 -- The l2 constraint was much more effective in our optimization scheme, and for us resulted in less noticeable perturbations. We also note that visual inspection by the authors concluded that nearly all adversarial directions found for the natural model were not discernible, and often the first 20 for the robust model were not discernible (on CIFAR, on MNIST this is less true because the original images are nearly binary).
>
> _Edit: Updated references; qualified notes on visibility of adversarial perturbations, as closer inspection revealed a small number of visible perturbations._

---

### Official Review · Reviewer_Pjpz · 2021-10-30

**Correctness:** 4
**Technical Novelty And Significance:** 3
**Empirical Novelty And Significance:** 3
**Recommendation:** 6
**Confidence:** 4

**Details Of Ethics Concerns:**

No obvious ethics concerns are identified for this paper.

**Main Review:**

Strong points:
1. The study of an interesting problem.

2. Some of the findings are not new but do have some insights.


Weak points:
1. The paper is poorly presented. It is hard to see the implications of the findings to DNN robustness. The ideas are poorly motivated.

2. There are so many different ways of studying the adversarial subspaces, why do the authors choose to study geometry in certain ways? How does this work help deep learning differently from prior works?

3. The findings in this work may not be completely new, some of them are caused by the smoothing effect of adversarial training.

4. The dimensionality of inputs/representation space has been thoroughly studied in existing works [1,2]. How the dimensionality investigation in this work is different? Is there a difference in studying the input space vs the representation space? It seems to me some of the studies make more sense in the representation space.

5. The geometric property of adversarial subspaces has been studied in [3], one well-known adversarial subspace characterization work.  What is the difference between this paper to [3], are the subspaces studied in [3] the same as here? In [3], they define adversarial subspaces to be the space around adversarial examples, which is quite intuitive. The adversarial subspaces here are defined to be where misclassification occurs (more general, not tied to the worse-case adversarial examples). Does this mean misclassificaiton=adversarial? How about incorrectly classified natural examples, i.e., natural generalization error?

6. Many understanding and adversarial defense work also provide important understandings about adversarial vulnerability, adversarial robustness and properties of adversarially-trained models. For example, the loss landscape understanding [4,5], decision tiltiling understanding [7], boundary smoothing [8] all look quite relevant. Without a thorough discussion of the existing understandings, it is hard to assess the novelty of this work and how the findings add to the adversarial robustness.

7. It is unclear how the understandings developed in this paper can help future research. Can it be used to design or train more robust DNNs, evaluate their robustness, or maybe just a visualization method to help better understand the model?

[1] First-order Adversarial Vulnerability of Neural Networks and Input Dimension, ICML 2019.

[2] The vulnerability of learning to adversarial perturbation increases with intrinsic dimensionality, WIFS, 2017.

[3] Characterizing adversarial subspaces using local intrinsic dimensionality, Ma et al. ICLR 2018.

[4] Interpreting and Evaluating Neural Network Robustness, IJCAI, 19.

[5] Adversarial Weight Perturbation Helps Robust Generalization, NeurIPS, 2020.

[7] A Boundary Tilting Persepective on the Phenomenon of Adversarial Examples.

[8] Theoretically Principled Trade-off between Robustness and Accuracy. ICML 2019.


**Summary Of The Paper:**

This paper studies the geometry of adversarial subspaces around input samples, with a focus on their spatial properties and relation to the decision boundary. Orthogonal adversarial directions are found by untargeted adversarial perturbation and are used to define the adversarial subspaces. The principal curvature decomposition is applied to find the principle curvature (the degree of linearity) of the decision boundary in adversarial supaces. Based on the results obtained from these experiments, the authors draw several conclusions including 1) input samples lie close to decision boundary in a large subspace; 2) the distance to the boundary grows with the increase of the subspace dimensionality; 3) boundary in more curved in adversarial subspaces and adversarial training tends to even the boundary to different classes.

**Summary Of The Review:**

This paper raises more questions than answers. It is hard to say the findings are reliable, accurate, or novel. The observations for adversarially trained models are more like a side effect of boundary smoothing, which is what adv training is designed for. Min-max optimization gives us a better explanation [8].

---

> ### Author Response · Authors · 2021-11-12
> **Paper significance & comparison to other studies (pt 1)**
>
> Dear reviewer Pjpz,
>
> Thank you for taking the time to review our submission. We appreciate the additional related citations that you have provided and your recommendation to improve the clarity of the paper. We would like to address each of your concerns, and we note that the updated manuscript integrates the clarifications so that our contributions and the context to related work is clear to future readers.
>
> 1 -- *The study is not sufficiently motivated:* Our method is entirely unique from other studies in that ours is the only work that we know of which looks at >1-dimensional spaces defined by adversarial directions. All other studies use random orthogonal perturbations, nearest clean neighbors, or alternative clean image directions to understand the decision boundary. This is important because we are able to observe and measure properties of the decision boundary where it is closest to test images. Other adversarial studies do this, but only at a single boundary point. We have demonstrated non-trivial behavior when one looks at more than one boundary point.
>
> Additionally, our study is the first to derive the equations for measuring the level-set curvature of the decision boundary (Appendix A).  We added the Appendix G to detail how all other previous works computed incorrect quantities. This curvature is important, as it speaks to the linearity of the boundary in the input space, which is something that has been highly relevant for many explanations for adversarial robustness.
>
> 2 -- *Why do the authors choose to study geometry in certain ways? How does this work help deep learning differently from prior works?* Together, our methods allow us to report the first assessment of the geometry of the actual boundary in >1 dimensions where it is closest to test set images. While there are many geometric quantities one could report, we decided that the distance to the boundary and the curvature of the boundary were fundamental prerequisites to any others, and thus chose those as our primary measurements.
>
> *Some of the findings are caused by the smoothing effect of adversarial training.* Our findings are congruent with the hypothesis that adversarial training smooths decision boundaries, which we support with unique quantitative measures. We now mention this in the new conclusion paragraph.
>
> 3 -- *How is the dimensionality investigation in this work different from existing works studying the inputs/representation space?* We are not measuring the dimensionality of the dataset manifold or of the input space itself. Rather, we are measuring the dimensionality of the subspace where the decision boundary is close to test set images. The decision boundary itself lives in the input space, so we appreciate you bringing up this important distinction. We do repeat our experiments on datasets with different input dimensionality (MNIST vs CIFAR). The adversarial subspace sizes and perturbation lengths were scaled in a predictable way, but we found that the relative trends between adversarial and natural training were largely the same. This suggests that, unlike robustness itself (Simon-Gabriel et al., 2019), they are not dependent on input dimensionality. We added a comment on this in Section 4 and Appendix D.
>
> 4 -- *The geometric property of adversarial subspaces has been studied in another well-known adversarial subspace characterization work.* Work from Ma et al. (2018) measured the local intrinsic dimensionality (LID) of adversarial subspaces, which indirectly estimates adversarial subspace dimensionality by measuring the distance from an adversarial example to a batch of nearest unperturbed dataset examples. This idea has been extended to algorithms for detecting adversarial examples (Ma et al., 2018; Mao et al., 2020), although Athalyeet al. (2018) generated adversarial examples with indistinguishable LID scores from natural examples. In terms of understanding ANN classifiers, such an approach leaves open the questions of how proximal the decision boundary is from input samples in greater than one dimension or the actual dimensionality of adversarial subspaces. Our approach is the first to pursue a more global examination of the decision boundary where it is nearest to test samples by defining adversarial subspaces using multiple orthogonal adversarial directions.
>
> 5 -- *The adversarial subspaces here are defined to be where misclassification occurs.* Thank you for bringing this to our attention. The experiment was repeated on 100 test images that were chosen to have equal label representation (10 images per label) and to be correctly classified by both networks. In other words, we are only interested in the decision boundary near correctly labeled inputs. We have added a description of the sampling and image selection procedure to the end of Section 2.1.
>
> _Edit: Updated references._

---

> ### Author Response · Authors · 2021-11-12
> **Paper significance & comparison to other studies (pt 2)**
>
> 6 -- *Without a thorough discussion of the existing understandings, it is hard to assess the novelty of this work and how the findings add to the adversarial robustness.* We have added a paragraph to the top of our conclusion section that puts our work into the context of other theories for adversarial vulnerability.
>
>
> 7 -- *It is unclear how the understandings developed in this paper can help future research.* We share your enthusiasm for finding applications of our findings. However, we chose instead to focus this paper on explaining the method and demonstrating what we found to be core quantities, namely boundary distance & curvature. We would point you to the final paragraph of our conclusion section, which lists a number of interesting applications, including: improving neural network performance with semi-supervised relabeling, architecture design, and adversarial robustness. We also note that our method is an interpretability approach that is quantitative and falsifiable, which has recently been argued as necessary to apply deep networks to sensitive industries, such as healthcare.
>
> We would appreciate it if you would let us know whether the above clarifies your concerns and changes your assessment of the impact of our work.

---

> > ### Comment · Reviewer_Pjpz · 2021-11-22
> > **Thanks for the clarification**
> >
> > Thanks to the authors for providing the detailed clarification. I went through the paper again and make sure I have understood the proposed tool correctly. With the new explanations, I realized that I was trying to interpret the tool from the adversarial example/attack perspective, however, the proposed is a general tool for studying the decision boundary geometry. Although it uses *adversarial directions* and is named as *adversarial subspaces*, it is more general. I hope the authors could make it clear at the beginning (abstract or intro) that this is not a study of adversarial examples nor worse case adversarial perturbations, but a general tool for characterizing the decision boundary. Otherwise, readers from an AML background might get confused.
> >
> > I am willing to increase my rating to 6.

---

> > ### Comment · Reviewer_Pjpz · 2021-11-22
> > **Another suggestion**
> >
> > The authors may also consider changing the title to something like *Adversarial Geometry of Deep Decision Boundary*. Just a thought.

---

### Official Review · Reviewer_Dw9Y · 2021-11-03

**Correctness:** 4
**Technical Novelty And Significance:** 3
**Empirical Novelty And Significance:** 3
**Recommendation:** 6
**Confidence:** 4

**Main Review:**

<Updated after the author response: Thank you for the detailed response, clarifying comments, and additional experiments. I have increased my score to 6.>

1. Using adversarial robustness to explain how and why deep networks perform well is an interesting idea, but it wasn't clear to me if this method can be used to explain the performance differences between a large array of standard deep networks. For example, if ResNet152 works better than AlexNet, can we use approaches introduced in this paper to explain their performances, or is the approach here limited to the comparison between adversarially trained vs. vanilla networks only? Empirical demonstration of the proposed approaches on standard deep networks would be helpful, as well as their limits and generalities.

2. There are several prior works which deal with the geometry and curvature of deep network representations and boundary surfaces [1,2]. How does this work differ from them?


3. One of the well-known inductive biases for adversarial robustness is to introduce noise or random perturbations [3,4]. Can you still apply the methods introduced here in the presence of additive noise or representation stochasticity?

4. The empirical findings here (on the increase in boundary distance within adversarial subspaces, and the decrease in boundary curvature in an adversarially trained model) are perhaps not too surprising. The results from the comparison between adversarially trained model vs. standard model are reassuring, but the paper can be made stronger by comparing several adversarially trained models (or several adversarial attack methods), and demonstrate that the reported trends hold against multiple adversarially robust models.

References

[1] Lahiri et al: https://arxiv.org/pdf/1606.05340.pdf

[2] Cohen et al: https://www.nature.com/articles/s41467-020-14578-5

[3] Li et al https://arxiv.org/abs/1809.03113

[4] Dapello et al https://proceedings.neurips.cc/paper/2020/hash/98b17f068d5d9b7668e19fb8ae470841-Abstract.html

**Summary Of The Paper:**

This paper proposes methods to understand the geometry of decision boundaries of ANN classifiers by using adversarial perturbations as a tool. They introduce two methods: (1) a method for finding adversarial subspaces (where input sample has minimal distance to the decision boundary); (2)  a method for measuring the curvature of a decision boundary. These methods allow for new findings such as (1) distance to the boundary grows with increasing dimensionality of adversarial subspace; and (2) decision boundary is more curved in adversarial subspace than random subspace; and (3) insights into why adversarial training works well.

**Summary Of The Review:**

Investigating the robustness of deep networks using the geometry of decision boundaries and adversarial subspaces is a well motivated and interesting idea. My main reservation about this paper is that the proposed interpretability metrics are demonstrated in a sanity-check style manner on only a small set of models. In my view, the utility of such interpretability metrics can be really demonstrated when they are applied against several models of similar types (e.g., an array of standard deep network models, and an array of adversarially robust models), and when it is shown that the empirical claims with the proposed metrics hold for a large array of standard models/training methods.

---

> ### Author Response · Authors · 2021-11-15
> **New results on an additional CIFAR architecture**
>
> Dear reviewer Dw9Y,
>
> Thanks for taking your time to review our submission; your proposed citations and empirical experiments were very helpful. We would like to address your main reservation and will follow up with a response to your remaining points in the coming days.
>
> You stated that your main reservation was “​​that the proposed interpretability metrics are demonstrated in a sanity-check style manner on only a small set of models.” Per your suggestion, we repeated our experiments on the state of the art WideResNet (WRN-70-16) architecture with Swish activations, from Gowal et al. [2020](https://arxiv.org/abs/2010.03593), with and without adversarial training on CIFAR. This allows for comparisons across architectures and training modality for a single dataset. The following links contain figures for the WRN-70-16-Swish network, and we have noted the corresponding PDF figure to compare against a RN-50-ELU architecture.
>
> * [https://ibb.co/L09xLxj](https://ibb.co/L09xLxj) -- Figure 2
> * [https://ibb.co/X4jrpMc](https://ibb.co/X4jrpMc) -- Figure 7
> * [https://ibb.co/rtDDDQZ](https://ibb.co/rtDDDQZ) -- Figure 8
>
> Because of computational limitations, we show the first 20 adversarial subspaces (instead of 50) across 86 images (instead of 100) and we do not include the curvature analysis. The full set of results will hopefully be ready before the review period ends, and certainly in time for the camera ready version. We will also make figures that directly compare the two architectures within a given training regime for the Appendix (see Appendix C). We found little change in the results, suggesting that the differences among architectures is much less significant than the difference between training schemes.
>
> As another control, we repeated the experiment for Figure 18 (measuring the length of the adversarial vector as one increases dimensionality on the MNIST trained networks) across five different seeds per training type. You can see the results via this link [https://ibb.co/NWLFdZr](https://ibb.co/NWLFdZr), or Figure 23.
>
> Finally, we would also point you to Appendix D, where our method illuminates some interesting differences between MNIST and CIFAR trained networks.
>
> In general, we agree with your desire to see the method run on a broad class of architectures, datasets, or training methodologies. However, extracting a 50-dimensional adversarial subspace on the CIFAR trained ResNet-50 from He et al. [2016](https://ieeexplore.ieee.org/document/7780459) takes about a half day per image on a single GPU, which then has to be followed by the curvature analysis. Thus, strictly due to computational constraints, we are reserving a large-scale empirical analysis for future work.
>
> In another reply we will follow up with a discussion on related work, noise robustness, and the broader novelty of our work. In the interim, we welcome any additional questions or comments.
>
> _Edit: Updated figure and appendix references_

---

> ### Author Response · Authors · 2021-11-18
> **Direct comparisons to new CIFAR architecture & addressing remaining comments**
>
> Dear reviewer Dw9Y,
>
> We would like to update you with some comparisons for the WRN-70 and RN-50 networks, as well as answers to the remaining comments in your original review. You can look at these figures [https://ibb.co/JKBqFZg](https://ibb.co/JKBqFZg); and [https://ibb.co/nB87wQw](https://ibb.co/nB87wQw) to see a direct comparison between the WRN and RN networks with adversarial and natural training, respectively. We show every other adversarial subspace (i.e. with dimensions 1, 3, 5, …, 19)  for readability. At the level of these aggregate statistics, the hyperparameter differences in adversarial training (red data) have more of an effect than the model architecture differences themselves (blue data), although all differences are much less pronounced than the differences between adversarially and naturally trained networks (red vs blue).
>
> 1-- and 4-- Inclusion of the state-of-the-art network trained by Gowal et al. [2020](https://arxiv.org/abs/2010.03593) allows us to directly compare architectures. Our new figures show that the reported trends do indeed hold against multiple adversarially robust models, although some interesting differences are present. They appear to largely hold across datasets as well, as shown in Appendix D.
>
> Our choice of adversarial attack method was constrained by our unique requirement to extract >1 orthogonal, untargeted adversarial directions. Outside of minor adjustments (such as the chosen p-norm), all other attack methods we considered include assumptions that would bias our found directions. We performed a large review of attack methods when we were piloting this study and we were unable to find others that suited our constraints. We do believe there are alternative objectives to be found, but we suspect this will be a novel contribution in itself.
>
>
> 2 -- Thank you for pointing us to these related works. We have included a detailed discussion on Poole, Lahiri, et al. ([2016](https://arxiv.org/pdf/1606.05340.pdf)) in Appendix G. The quick answer is that while their analysis is similar to ours in spirit, it is limited to low-dimensional (ring) stimuli and networks with specific (mean-zero Gaussian) weight & bias structure. They do not look at trained neural networks. The work of Cohen et al. ([2020](https://www.nature.com/articles/s41467-020-14578-5)) measures the geometry of the neuron response manifold as one increases layer depth, although they do not look at adversarial directions, principal curvatures, or at decision boundary curvature.
>
> 3 -- Measuring boundary curvature for non-smooth (e.g. ReLU) and stochastic networks is something that has been top-of-mind for us as well. Doing so requires a costly Hessian approximation step. This can be done via finite-differences approximation, as in Moosavi-Dezfooli et al. ([2019](https://openaccess.thecvf.com/content_CVPR_2019/papers/Moosavi-Dezfooli_Robustness_via_Curvature_Regularization_and_Vice_Versa_CVPR_2019_paper.pdf)), Eq 1 and the surrounding discussion. We are developing an optimization-based approach that we expect to be more accurate and faster than the Moosavi-Dezfooli et al. method, which will allow us to scale to analysis on VOneNet, for example. However, the additional technical challenges and computational difficulty have led us to decide to leave this for future work, and focus the present study on showcasing the ideas using twice-differentiable neural networks.
>
> We would appreciate it if you would let us know whether the above clarifies your concerns and changes your assessment of the impact of our work.
>
> _Edit: Updated figure and appendix references_

---

> ### Author Response · Authors · 2021-11-26
> **WRN-70 curvature results**
>
> Dear reviewer Dw9Y,
>
> As a quick follow-up, please find the curvature results and adversarial class compositions via these links [https://ibb.co/vmkJqCF](https://ibb.co/vmkJqCF) and [https://ibb.co/RyRCtD2](https://ibb.co/RyRCtD2), which should be compared against Figures 4 and 5, respectively. The curvature analysis was conducted with 10 test images and 8-dimensional subspaces, although we will increase these numbers before the camera-ready deadline. The results are largely consistent with those for the RN-50 network.

---

### Official Review · Reviewer_kBnb · 2021-11-04

**Correctness:** 3
**Technical Novelty And Significance:** 2
**Empirical Novelty And Significance:** 3
**Recommendation:** 6
**Confidence:** 4

**Main Review:**

The paper studies the curvature properties of class boundaries of
neural network architectures. A modified version of the optimization
strategy for adversarial perturbation vector generation (Szegedy 2014)
is used to obtain an adversarial subspace. A distribution of the
distance of decision boundary (equi-logit points) to the test samples
is computed in the adversarial subspace. The distribution shows that
the distances grow slowly as larger adversarial subspaces are
considered. This explains the success of adversarial attacks. The
space operator is next used to obtain the tangent subspace. Basis of
the tangent subspace is ordered based on curvature. The subspace pull
back is defined to measure the susceptibility of the image to an
attack. It is observed that the subspace pull back distribution is
supported by most of the directions.

It is also observed that use of adversarial training increases the
entropy of the class distribution close to a boundary. It also
modifies the pull back factor. This is a viable explanation of the
success of adversarial training strategies.

The paper uses a number of known analysis techniques to an elegant
explanation of adversarial robustness. Empirical studies support the
claims.

**Summary Of The Paper:**

The paper studies the curvature properties of class boundaries of
neural network architectures. A modified version of the optimization
strategy for adversarial perturbation vector generation (Szegedy 2014)
is used to obtain an adversarial subspace.
The paper uses a number of known analysis techniques to an elegant
explanation of adversarial robustness. Empirical studies support the
claims.



**Summary Of The Review:**

I'm leaning towards accepting this paper because it conducts an extensive analysis with existing approaches towards elegant explanations of adversarial robustness,

---

> ### Author Response · Authors · 2021-11-11
> **novelty of analysis techniques**
>
> Dear reviewer kBnb,
>
> Thank you for taking the time to review our submission. We are glad that you found the explanations elegant and that the main takeaways were clear to you. We would like to address your primary comment, that “the paper uses a number of known analysis techniques”, as we believe this may be due to a misunderstanding that could very well be shared by other readers. While we did build on known techniques, we had to extend or fix all of them to make them applicable to our study. We will make the novelty of our techniques more explicit in our updated manuscript. Here is a summary:
>
> *Adversarial decomposition:* The underlying objective of our optimization procedure is similar to that proposed by Szegedy et al. (2014) for finding a single adversarial direction, but several additional components were necessary to successfully find an orthogonal set of untargeted vectors. Namely, the additional constraints on the objective in Equation (1), and the binary search procedure wrapped by the interior-point (Ipopt) optimizer. Other previous works (Tramer, 2017; He, 2018) have attempted to find orthogonal directions pointing to the decision boundary near individual adversarial images, which is different from our method of finding orthogonal adversarial directions centered on a test image. Additionally, in Appendix F we explain how those two methods are flawed in providing proper geometric subspaces because they use clipping operations, which impacts one’s ability to interpret their results in a meaningful way. Our method avoids clipping by explicitly considering the boundedness of the input domain during optimization. Thus, to the best of our knowledge, our study is the first to find a set of orthogonal adversarial vectors for a given input image, which was a necessary prerequisite for most of the empirical results presented.
>
> *Curvature analysis:* While previous work has utilized Riemannian curvature analysis to understand neural networks (Fawzi, 2016; Poole, 2016; Moosavi, 2018, 2019; Golden, 2019), each study measures different quantities without complete details of the derived formulas. To demonstrate the validity of our own approach, we reproduced their calculations and measured curvature on n-dimensional spherical and hyperboloid manifolds. We found the measurements using their formulas to be off by significant margins (see Appendix G). As such, one of our major contributions to this literature is a rigorous derivation and discussion of these curvature quantities, which we provide in section 2.2 and Appendix A. This leads to what we believe is the first principled and analytically supported technique for measuring the decision boundary curvature for trained nonlinear neural networks. We will add an appendix section (Appendix G) to demonstrate the comparison to alternative curvature methods and we will also publish the code to reproduce our findings, including these comparison tests.
>
> We would appreciate it if you would let us know whether the above clarifies the novelty of our analysis and changes your assessment of the significance of our work.
>
> _Edit: Updated figure and appendix references._

---

### Author Response · Authors · 2021-11-22
**Summary of revisions**

We would like to thank all reviewers for their valuable inputs and constructive feedback, summarizing our work as “elegant” (kBnb), “well motivated” (Dw9Y), “interesting” (Dw9Y, Pjpz, xfHm), and “principled” (xfHm). Their suggestions have led to a number of exciting new results and updates, which we think considerably improves the quality of the submission. Namely:

- *Additional architecture experiments:* (Dw9Y, xfHm) We reran all of our experiments on the state of the art WideResNet-70 architecture with Swish activations from Gowal et al. [2020](https://arxiv.org/abs/2010.03593). We plotted direct comparisons in Appendix C. This together with the MNIST results in Appendix D demonstrates that our observed trends hold across datasets and architectures. The curvature analysis for this network is still computing, although we anticipate that it will be completed before the end of the review period, and certainly before the camera-ready deadline.

- *Sanity checks and controls:* (xfHm) We ran four new experiments for checking the validity of our results across training and sampling seeds, which can be found in Appendix E.

- *Comparisons to previous work:* (Pjpz, Dw9Y, xfHm) We added 21 new cited works to clarify the novelty of our study and more properly place our work in the context of related studies in the field. We expanded on a paragraph now placed in our conclusion section that gives context for our work relative to current theories of adversarial susceptibility. We added Appendix G, which gives a full exposition of the relationship between our work and other methods for computing decision boundary curvature. Appendix F also discusses our relationship to other studies for measuring the dimensionality of adversarial subspaces.

- *Clarity of exposition:* (kBnb, Pjpz) We added or modified phrasing throughout the text to make our contributions more explicit and to provide better context for our findings. We now highlight that our analysis is the first to be conducted with >1 adversarial directions and the first to provide detailed derivations supporting curvature calculations.  Thus we provide the first geometric description of the subspace near test-set images where the decision boundary is closest.

---

> ### Author Response · Authors · 2021-11-26
> **Final results**
>
> Dear reviewers and area chair,
>
> We would like to update you all with the curvature and class distribution plots for the WRN-70 network. The new figures can be found via [https://ibb.co/vmkJqCF](https://ibb.co/vmkJqCF) and [https://ibb.co/RyRCtD2](https://ibb.co/RyRCtD2), and should be compared against Figures 4 and 5, respectively. The curvature analysis was conducted with 10 test images and 8-dimensional subspaces due to computational restrictions. We will increase these numbers before the camera-ready deadline. We also repeated the entropy calculations and found that the robust network still demonstrated larger entropy of unique proximal adversarial labels per test image class (as was shown for the RN-50 network in Table 1).
>
> With these final results, we believe we have addressed all of the reviewer concerns. We have not received replies to all of our responses, but we hope that the additional experiments and clarifications will result in an improved score from the remaining reviewers. If there are any additional concerns, we would greatly appreciate an opportunity to continue the discussion.

---

### Comment · Area_Chair_PjZo · 2021-11-26
**Further questions**

I have some additional questions/concerns that were not addressed by the current reviews. I would appreciate if either reviewers or the authors could answer these questions.

1. It doesn't make sense to me that the authors claim the "adversarial subspace" is < 50 dimensional for the considered problems.

My understanding of the literature is that there are simple experiments which demonstrate this space of errors is definitively full rank (the dimension of the input space). For example, one can generate adversarial inputs x_adv which remain adversarial when random gaussian noise is added to the input. This can only be possible if x_adv lies in a local region that spans all dimensions of the input space, as the measure of a rank K < N subspace of R^N is measure 0.

More generally [1, 3] show that "adversarial examples" are nothing more than the boundary points of the error set the rest of the ML community refers to as "out-of-distribution generalization error". E.g. errors found by randomly perturbing an input with Gaussian noise (as in this benchmark [2]) and adversarial exampels lie in the same connected region of the error set (see [1] Figure 3). In this sense, the Cifar-10-C benchmark measures the volume of the "adversarial subspace" by randomly perturbing inputs and reporting generalization error. This again shows that this space has measure > 0 which can only occur if it were full rank.

2. Related to my first conern, I'm concerned the procedure the authors define in equation (1) doesn't satisfy some sanity checks one would hope a geometric measure of subspace dimensionality should satisfy. For example, suppose the error set in R^n was defined as all inputs which satisfy x_1 >= (x_2^2 + ... + x_n^2) + 1 --- that is a generalization of the quadratic y > x^2 + 1. If we were to run algorithm (1) starting at x = (0, 0, ....). Our first vector would simply be d_1 = (1, 0, 0, 0) which happens to be the normal vector to the error set. After which the algorithm would terminate b/c all orthogonal directions to d_1 do not intersect the error set, thus we would report the subspace in this example were dimensionality 1 which is not correct.



1. https://arxiv.org/pdf/1901.10513.pdf
2. https://arxiv.org/abs/1903.12261
3. https://slideslive.com/38930579/why-adversarial-examples-feel-like-bugs

---

> ### Author Response · Authors · 2021-11-28
> **Addressing further questions**
>
> Dear Area Chair,
>
> The relationship between our work and out-of-distribution (ood) robustness is quite relevant, so thank you for bringing it up.
>
> First and foremost, we would like to clarify what we are measuring. We are not mapping out subspaces near adversarial images, i.e. we’re not answering the question “in how many directions can I go from an adversarial and stay adversarial?” As a number of previous studies have shown, and as you suggested, that space will be full rank in most cases. Instead, we are asking a question that was previously unanswered: “in how many directions can I go from a source image and reach the decision boundary quickly, i.e. find an adversarial?” or, more precisely, "how close (and how curved) is the decision boundary near source images in more than one unique direction?" Therefore, we do not look at the subspace around a given adversarially perturbed input, but instead we estimate distances to and curvature of the decision boundary in vulnerable subspaces around a given (clean/within-domain) input image by means of many orthogonal perturbations originating from that input.
>
> We find that answers to our question provide important complementary information to the above quoted alternative. The subspace around an adversarial is likely to be full rank and the behavior of the network in the vicinity of an existing adversarial reveals little more about the network’s ood robustness. On the other hand, our investigation reveals (to us) an interesting and original perspective. Namely, that the decision boundary is close to test images in many orthogonal directions, and that the distance to the boundary grows slowly and uniformly as one looks for additional directions. The continuity and dimensionality of close boundary regions cannot be known by measuring a sampling of ood images and reporting an ood error rate, or by exploring the space near an adversarially perturbed input. In other words, while [1, 3] have identified adversarial examples as the nearest points in the set of ood errors, no previous work has measured the distance to the ood error set in > 1 dimensions from a given input. Thus, our results are consistent with and expand upon the perspective presented by Gilmer and colleagues.
>
> In light of this, and to specifically address your first concern, we did not intend to give the impression that these subspaces of interest are <50 dimensional. In fact, we do not believe it is sensible to report a specific dimensionality. The only reason our plots extend to 50 dimensions is because we chose to stop the optimizer there (due to computational constraints). The observed trends suggest that the boundary will smoothly grow away from test inputs as one looks for additional (>50) dimensions.
>
> The above explanation rests on the assumption that our optimizer found a good solution -- that is to say that the found orthogonal vectors point in directions where the decision boundary is nearest to the clean image. That our results are consistent across images, models, seeds, and datasets gives us confidence that we are sufficiently extracting orthogonal adversarial perturbations for a given input image. Furthermore, our empirical findings of >= 50 adversarial perturbations for 100% of sampled images is reassuring of the validity of our approach. However, we readily admit that our approach is a greedy approximation and thus can be seen as an upper bound of estimates of distances to the decision boundary. This does not negatively impact our findings, since a better optimization would only lead to more drastic results.
>
> We think that your example of x_1 >= (x_2^2 + ... + x_n^2) + 1 can be taken as an additional illustration of how our approach can provide a more interesting answer than measuring the subspace around a given adversarial or the dimensionality of the error set. As you correctly point out, the error set has full rank in this case. However, the given example is very well-conditioned in the sense that in most directions from the origin the model decision won’t change at all, not to mention at close distances. Our method quantifies this property: there is only one orthogonal direction from the origin to flip the decision.

---

> ### Author Response · Authors · 2021-11-28
> **Proposed modifications**
>
> We propose the following modifications to our manuscript to clarify the distinctions made in the earlier reply.
>
> We see that the term adversarial subspaces might be misleading and we would be willing to change it to provide more clarity (this was also suggested by reviewer Pjpz). For example, we could change the title to: “The geometry of neural network decision boundaries”. We are open to considering other title suggestions.
>
> We suggest this modification to the abstract (text before and after the ellipses would be left unchanged):
>
> >[...] The primary contribution of this work is to further our understanding of the decision boundary geometry of ANN classifiers by utilizing such adversarial perturbations. Specifically, we find a set of orthogonal directions where the decision boundary lies closest to input samples. Our analysis reveals that this occurs in a large subspace, where the distance to the boundary grows smoothly and sub-linearly as one increases the dimensionality of the subspace. We undertake analysis to characterize the geometry of the boundary, which is more curved within the adversarial subspace than within a random subspace of equal dimensionality. [...]
>
> And adding these sentences to the introduction:
>
> >[...] Unlike all of the existing approaches to characterize decision boundaries in the input space, which rely on at most one adversarial direction per input, we propose jointly defining a subspace of adversarial directions. Ours is also an alternative approach to previous works that explore subspaces near individual adversarial examples. While they investigate generalization error near a single adversarial perturbation from a given input, we alternatively measure the proximity and curvature of the decision boundary near source images in more than one unique direction. The resulting analysis leads to a general explanation of classifier decision boundaries, as well as adversarial susceptibility, in the subspace where the boundary is closest to the input example. [...]
>
> And finally this rewrite of the relevant related works paragraph:
>
> >As opposed to looking at subspaces in the proximity of a clean (within-domain) image (which is what we’re doing here), several previous works have explored subspaces proximal to a given adversarial perturbation (e.g.  Fawzi et al., 2016; Tabacof & Valle, 2016; Liu et al., 2017; Tramèr et al. 2017; He et al., 2018). For example, Tramèr et al. (2017) and He et al. (2018) reported the dimensionality of a subspace proximal to adversarial examples where classification errors still exist. Fawzi et al. (2016) looked at the nearest error point to an image when constrained to search within random subspaces. Others used random perturbations to understand the classifier decision space near individual adversarial examples. These works provide a complementary perspective to our own by providing an understanding of the space of errors around individual non-robust examples. Our analysis is unique in that it allows us to characterize the decision boundary near clean images in more than one dimension, where we find a large set of unique adversarial directions. Consequently, we are the first to report the class, number, and proximity of orthogonal adversarial examples nearby clean inputs. A finer, but still important distinction is that most of the above studies perform a clipping operation after defining or finding the desired directions, which results in a non-orthogonal set. We overcome this problem by including the valid pixel range as inequality constraints in the optimization process (see Appendix F for a more detailed discussion). Work from Ma et al. (2018) measured the local intrinsic dimensionality (LID) of subspaces near adversarial examples, which indirectly estimates adversarial subspace dimensionality by measuring the distance from an adversarial example to a batch of nearest unperturbed dataset examples. This idea has been extended to algorithms for detecting adversarial examples (Mao et al., 2020), although Athalye et al. (2018) generated adversarial examples with indistinguishable LID scores from natural examples. In terms of understanding ANN classifiers, such an approach leaves open the question of how proximal the decision boundary is from input samples in greater than one dimension. Finally, our work is complementary to previous studies that investigate the relationship between adversarial robustness and input dimensionality (Amsaleg et al., 2017; Simon-Gabriel et al., 2019), since the comparative trends observed between robust and natural networks are relatively unchanged with small (MNIST) or large (CIFAR) dimensional inputs.

---

### Decision · Program_Chairs · 2022-01-20

**Decision:**

Reject

**Comment:**

The work investigates the decision boundary of neural networks by quantifying in various ways the shape and curvature of the error set local to correctly classified inputs, dubbed the "adversarial subspace". First, a method is introduced which seeks to find the largest set of orthogonal directions starting at in input x which will all intersect the error set local to an image. This is motivated as a certain geometric measure of the error set, large sprawling error sets may have many orthogonal directions which intersect it local to the given input while small narrow error sets may have relatively few. Using this geometric measure, the authors compare the shape of adversarial subspaces of various image models both with and without adversarial training. After the rebuttal period, reviewers all felt that the work was borderline, with no one strongly advocating for the work. As noted by some reviewers, while some experiments may be interesting it was unclear what new insights the work contributes. For example, the authors argue that the change in the geometry of the error set explains why adversarial training works. It is unclear how this is an explanation more than it is simply an observation that the error set geometry has changed. An analogy would be trying to explain why Resnet-50 performs better than AlexNet by showing that it has higher test accuracy---this only shows that it is better, but doesn't explain why.

During the discussion period the AC raised additional concerns regarding a sanity check that the author's main algorithm should pass. In particular, consider an error set x_1 >= K(x_2^2 + ... + x_n^2) + C, parameterized by constants K and C > 0. For all choices of K and C and starting point x = (0, ..., 0), the authors main algorithm will always return 1 as the dimensionality of this error set. It will find the vector (1, 0, ..., 0) and then terminate. However, this is problematic because we can choose K and C to make this error set either very narrow (e.g. K=C=100) or very wide (K = .000001, C = .00001)---the proposed algorithm will be unable to distinguish between this two extremes. Given this, it seems that greedily selecting the set of orthogonal directions starting at x can be very suboptimal if the intent is to find a maximum sized set of orthogonal directions.

To conclude, the work would be substantially improved if it addresses two major weaknesses. First, there needs to be a clearer motivation for studying this notion of geometry of the error set, what new insights can the authors provide other than adversarial training changes the shape of the error set? Second, the method doesn't seem to be principled given it is unable to distinguish between the two extreme cases discussed above.